

# Earth's future climate and its variability simulated at 9 km global resolution

Ja-Yeon Moon[1,2], Jan Streffing[3], Sun-Seon Lee[1,2], Tido Semmler[3,4], Miguel Andrés-Martínez[3], Jiao Chen[3], Eun-Byeoul Cho[1,2], Jung-Eun Chu[5], Christian Franzke[1,6], Jan P. Gärtner[3], Rohit Ghosh[3], Jan Hegewald[3,7], Songyee Hong[8], Nikolay Koldunov[3], June-Yi Lee[1,6], Zihao Lin[5], Chao Liu[9], Svetlana Loza[3], Wonsun Park[1,6], Woncheol Roh[1,2], Dmitry V. Sein[3,10], Sahil Sharma[1,2], Dmitry Sidorenko[3], Jun-Hyeok Son[1,2], Malte F. Stuecker[11], Qiang Wang[3], Gyuseok Yi[1,6], Martina Zapponini[3], Thomas Jung[3,12*], Axel Timmermann[1,2*]

[1]IBS Center for Climate Physics, Busan, 46241, Republic of Korea
[2]Pusan National University, Busan, 46241, Republic of Korea
[3]Alfred Wegener Institute, Helmholtz Centre for Polar and Marine Research, Bremerhaven, 27570, Germany
[4]Met Éireann, 65-67 Glasnevin Hill, D09 Y921, Dublin, Ireland
[5]Low-Carbon and Climate Impact Research Centre, School of Energy and Environment, City University of Hong Kong, Hong Kong, China
[6]Department of Climate System, Pusan National University, Busan, 46241, Republic of Korea
[7]Gauß-IT-Zentrum, Braunschweig University of Technology (GITZ), Braunschweig, Germany
[8]SSG services, Lenovo, Seoul, 06141, Republic of Korea
[9]Irreversible Climate Change Research Center, Yonsei University, Seoul, 03722, Republic of Korea
[10]Shirshov Institute of Oceanology, Russian Academy of Science, Moscow 117997, Russia
[11]Department of Oceanography and International Pacific Research Center, University of Hawaiʻi at Mānoa, Honolulu, 96822, USA
[12]Department of Physics and Electrical Engineering, University of Bremen, 28359, Bremen, Germany

*Correspondence to*: Thomas Jung (Thomas.Jung@awi.de) and Axel Timmermann (axel@ibsclimate.org)

**Abstract.** Earth's climate response to increasing greenhouse gas emissions occurs on a variety of spatial scales. To assess climate risks on regional scales and implement adaptation measures, policymakers and stakeholders often require climate change information on scales that are smaller (less than 10 km) than the typical resolution of global climate models [$O(100$ km$)$]. To close this important knowledge gap and consider the impact of small-scale processes on the global scale, we adopted a novel iterative global earth system modeling protocol. This protocol provides key information on Earth's future climate and its variability on storm-resolving scales (less than 10 km). To this end we used the coupled Earth system model OpenIFS-FESOM2 (AWI-CM3) with a 9 km atmospheric resolution (TCo1279) and a 4–25 km ocean resolution. We conducted a 20-year 1950 control simulation and four 10-year-long coupled transient simulations for the 2000s, 2030s, 2060s, and 2090s. These simulations were initialized from the trajectory of a coarser 31 km (TCo319) SSP5-8.5 transient greenhouse warming simulation of the coupled model with the same high-resolution ocean. Similar to the coarser resolution TCo319 transient simulation, the high resolution TCo1279 simulation with SSP5-8.5 scenario exhibits a strong warming response relative to





present-day conditions, reaching up to 6.5 °C by the end of the century at $CO_2$ levels of about 1,100 ppm. The TCo1279 high
resolution simulations show a substantial increase in regional information and granularity relative to the TCo319 experiment
(or any other lower resolution model), especially over topographically complex terrain. Examples of enhanced regional
information include projected changes in temperature, rainfall, winds, extreme events, tropical cyclones, and in the
hydroclimate teleconnection patterns of the El Niño-Southern Oscillation and the North Atlantic Oscillation. The novel
iterative modelling protocol, that facilitates storm-resolving global climate simulations for future climate time-slices, offers
major benefits over regional climate models. However, but it also has some drawbacks, such as initialization shocks and
resolution-dependent biases, which will be further discussed.

**1 Introduction**

Previous generations of global climate models have revealed fundamental insights into the large-scale response of the climate
system to past and future anthropogenic greenhouse forcing. To further provide crucial information on regional scales, several
international and domestic regional downscaling efforts have been launched, such as the Coordinated Regional Climate
Downscaling Experiment (CORDEX) (Fig. 1), which – depending on the domain of interest – can simulate climate features
down to scales of 8–25 km (Giorgi et al., 2012; Jacob et al., 2014; Giorgi and Gutowski, 2015; Gutowski et al., 2016). This
scale is of great interest to stakeholders who plan to assess risks of future climate change or implement specific climate change
adaptation measures (Lesnikowski et al., 2016; Pacchetti et al., 2021; Biswas and Rahman, 2023; Petzold et al., 2023; Jebeile,
2024). One of the disadvantages of regional model projections is that they use boundary condition input fields of coarser
resolution global models, which often do not properly resolve important mesoscale processes, such as tropical cyclones.
Another modelling approach that was pursued previously are pseudo-global warming experiments, in which coarser-resolution
Coupled Model Intercomparison Project (CMIP)-based SST patterns were used to force high-resolution atmospheric models
with resolutions down to 8 km (Jung et al., 2012; Kinter et al., 2013). As of recently, however, running global fully coupled
earth system models on scales of regional models has been prohibitively expensive and beyond the capability of many
supercomputers (Hohenegger et al., 2023; Rackow et al., 2024). Only in the last two years examples of seasonal (Hohenegger
et al., 2023) and multi-year (Rackow et al., 2024) simulations of coupled kilometer scale climate models have been made
available.



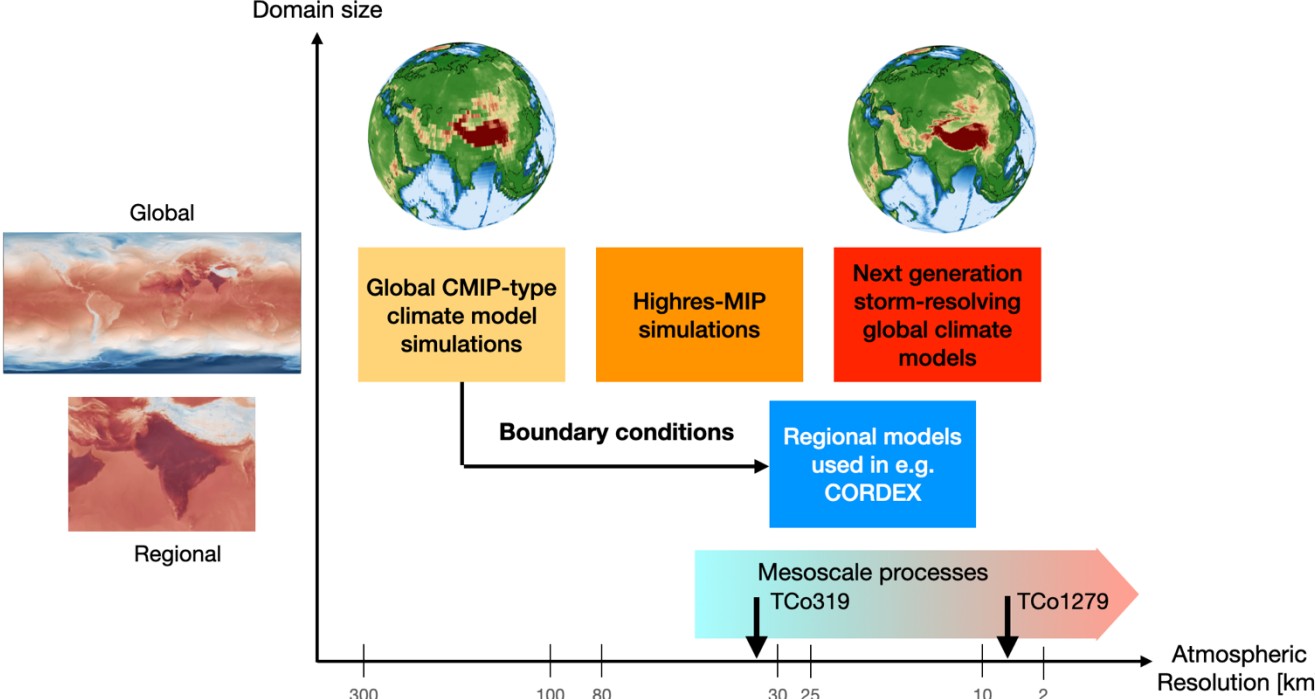

**Figure 1: Schematics, illustrating the modelling hierarchy of global and regional earth system models used for future climate change projections. Model simulations in this study are based on the OpenIFS-FESOM2 (AWI-CM3) model, which uses TCo319 and TCo1279 (cubic-octahedral spectral truncation) at 31 and 9 km, respectively. Our TCo1279 global model simulation employs a horizontal resolution that is similar to that used in regional models, such as in CORDEX simulations. The atmosphere and ocean model set-ups and resolutions for the TCo319 and TCo1279 configuration of the AWI-CM3 model are depicted in Figs. 2, 3.**

One of the first coordinated efforts to conduct coupled greenhouse warming simulations until 2050 CE with resolutions higher than the ones used in global earth system models, which participated in the CMIP Phase 6 (Eyring et al., 2016; Danabasoglu et al., 2020; Meehl et al., 2020), is the HighResMIP modelling project (Haarsma et al., 2016) (Fig. 1). In HighResMIP and other related global coupled modelling efforts (Chang et al., 2020; Huang et al., 2021), earth system models adopted atmospheric resolutions of about 25-km or larger. This higher resolution perspective on climate change has revised substantially our understanding of key climate processes and their sensitivity to various types of forcings, including the El Niño-Southern Oscillation (Wengel et al., 2021), tropical cyclones (Vecchi et al., 2019; Chu et al., 2020; Raavi et al., 2023), the East Asian Summer Monsoon (Liu et al., 2023), or atmospheric rivers (Nellikkattil et al., 2023). Still climate models run at this resolution or finer (Hohenegger et al., 2023; Rackow et al., 2024) require extensive computing resources, which limits the number and length of model simulations that can be conducted, including even test simulations, spin-ups or optimization runs that need to be performed to obtain a realistic modern climate mean state.





To close the resolution gap between regional climate models that only cover specific geographic domains and global climate

models, it is necessary to use modelling systems that are highly scalable on high performance computing systems and that can

be readily configured in different resolutions. To this end, we chose to conduct greenhouse warming simulations with the

OpenIFS/FESOM2 (AWI-CM3) model (Fig. 2), which has been adopted in recent studies in atmospheric resolutions of 100

86  km, 61 km, and 31 km (Streffing et al., 2022; Shi et al., 2023). The AWI-CM3 model uses the ECMWF IFS TCo atmospheric

grids. For this grid type the theoretically infinite spectral space is truncated to truncation number T, and in grid point space a

cubic octahedral (Barker et al., 2020) reduced Gaussian grid with four grid points sampling the smallest spherical harmonic is

used (Malardel et al., 2015). The IFS numerical implementation of the hydrostatic dynamical core scales extremely well on

HPC systems (Table 1) and serves as one of the principal enablers of the work we present here.

The primary goals of our study are 1) to identify the fidelity of the high resolution model and its biases under present-day

conditions, 2) to simulate future climate change with a state-of-the art coupled earth system model at atmospheric scales of 9

94  km and comparable ocean scales, 3) to provide key information on the regional aspects of mean changes in temperature,

precipitation, wind, and 4) to further document shifts in extreme events and modes of climate variability, such as the Madden

Julian Oscillation (MJO), the El Niño-Southern Oscillation (ENSO), and the North Atlantic Oscillation (NAO). This is

achieved by first conducting a transient SSP5-8.5 simulation at lower atmosphere resolution (31 km, TCo319), but with the

same ocean resolution and branching off 13-year and 10-year-long coupled simulations with the higher atmosphere resolution

(9 km, TCo1279) for years 2000s and 2030s, 2060s, 2090s (Fig. 3). In addition, a 20-year-long control simulation is conducted

00  at high resolution which uses constant 1950 greenhouse gas and aerosol conditions. We also repeated the 2090s chunk on the

01  Korea Meteorological Administration (KMA) supercomputer GURU. The results from this additional experiment are only

02  used in section 6 to extend the dataset that is used for the analysis of changes in extreme events, climate variability and

03  teleconnections. To the best of our knowledge, these new simulations are the highest resolution fully coupled global

04  simulations of future climate change reaching 2100 CE conducted to date.

05

06  The paper is organized as follows: in section 2 we will introduce the model set-up and AWI-CM3 performance for TCo319

07  and TCo1279. Section 3 provides an overview of the high-resolution present climate simulations in terms of both atmospheric

08  and oceanic processes. Section 4 describes the sensitivity of the model to future climate change and presents global-mean-

09  temperature-normalized climate change patterns, and section 5 focuses on the present and future statistics of atmospheric

extreme events. Our manuscript further emphasizes the impact of modes of natural climate variability on regional climates and

how these impact/teleconnection patterns may change in future (section 6). Section 7 concludes with a summary and

discussion.



While this manuscript features new scientific results on the high-resolution response of the earth climate system to greenhouse

warming, it also proposes a novel protocol for high-resolution coupled simulations and provides a reference for TCo1279

AWI-CM3 simulations. The data access links are shared in the corresponding section.

**2 Model Description and Experimental set-up**

Our study is based on the AWI-CM3 coupled climate model (Streffing et al., 2022), which employs the OpenIFS atmosphere

(cycle 43r3) (Huijnen et al., 2022; Bouvier et al., 2024; Savita et al., 2024) with hydrostatic approximation, the WAM surface

gravity wave model (Komen et al., 1996) and the hydrology model H-TESSEL (Balsamo et al., 2009) (Fig. 2). The ocean

model used is the Finite volumE Sea ice- Ocean Model (FESOM2) (Danilov et al., 2017; Koldunov et al., 2019a; Scholz et

al., 2019), which also includes the FESIM sea-ice module (Danilov et al., 2015). The model components are communicating

with each other via the OASIS3-MCT coupler (Fig. 2) and the Runoff mapper. OpenIFS output is managed through a parallel

XIOS IO server. For the simulations described in this study, we choose two different model configurations – the "medium"

resolution set-up (from here on MR) with TCo319 atmosphere resolution (Fig. 3a), with grid spacing of about 31 km near the

equator and the largest at about 38 km at high latitudes. The ocean uses the FESOM2 DART mesh, which employs a spatially

variable resolution (Fig. 3c) to optimally resolve the regional Rossby radius of deformation in regions with high eddy activity

(Streffing et al., 2022). In the Arctic, the ocean grid size corresponds to about 5 km, which can resolve large-scale sea-ice

cracks (Supplementary Movies S1, S2) (Wekerle et al., 2013; Wang et al., 2018; Koldunov et al., 2019b; Moon et al., 2024b).

For the tropical ocean an average resolution of about 12–25 km is adopted, which allows for a reasonable representation of

tropical instability waves (Small et al., 2003; Holmes et al., 2019), and island effects (Eden and Timmermann, 2004)

(Supplementary Movie S3) (Moon et al., 2024b). Moreover, the DART mesh exhibits increased spatial resolution of about

7.5–10 km in coastal regions. This improves the representation of nearshore processes, such as upwelling, coastal eddies or

shelf interactions, which are often not properly resolved in lower-resolution climate models participating in CMIP.



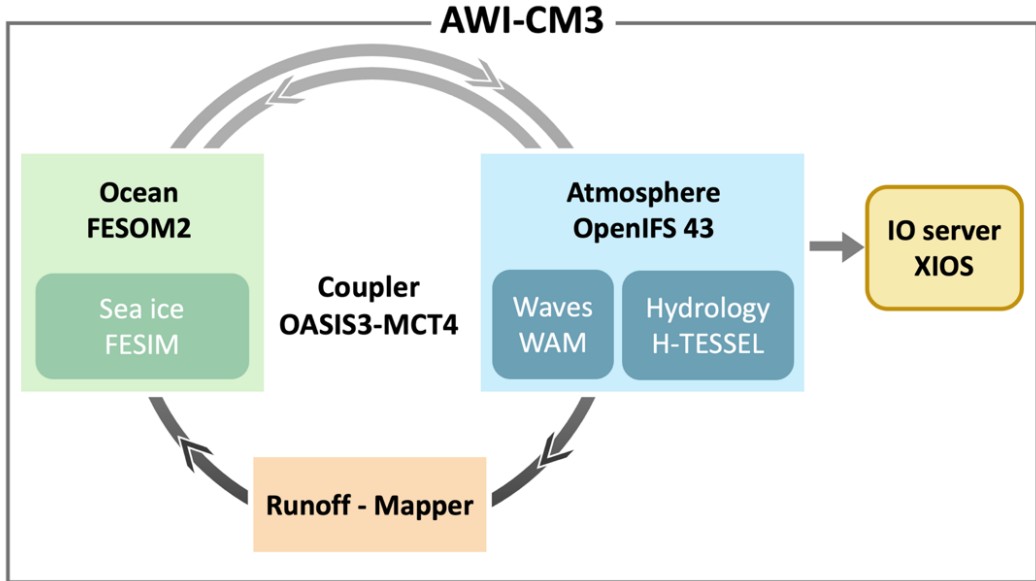

**Figure 2: Coupling schematics of AWI-CM3 showing the modelling subcomponents used for the MR and HR set-up.**

In addition to the MR case, we also run the AWI-CM3 model with a high resolution (hereafter referred to as HR) setup, which for OpenIFS corresponds to the TCo1279 atmospheric grid. This configuration attains the highest resolution of about 8 km near the equator and the lowest is about 10 km at high latitudes (Fig. 3b, right panel). This is the resolution realm that is normally adopted by regional climate models and current operational weather forecast models (Fig. 1). It is suitable to represent the regional features of tropical cyclones and capture the topographic details of prominent mountain ranges on our planet. The MR and HR configurations both use the same DART 80-layer ocean mesh and 137 pressure levels in the atmosphere extending from the surface to 0.01 hPa in the upper stratosphere. In both configurations the stratosphere can be considered sufficiently well-resolved for representing sudden stratospheric warming events, but not the stratospheric Quasi-Biennial Oscillation.





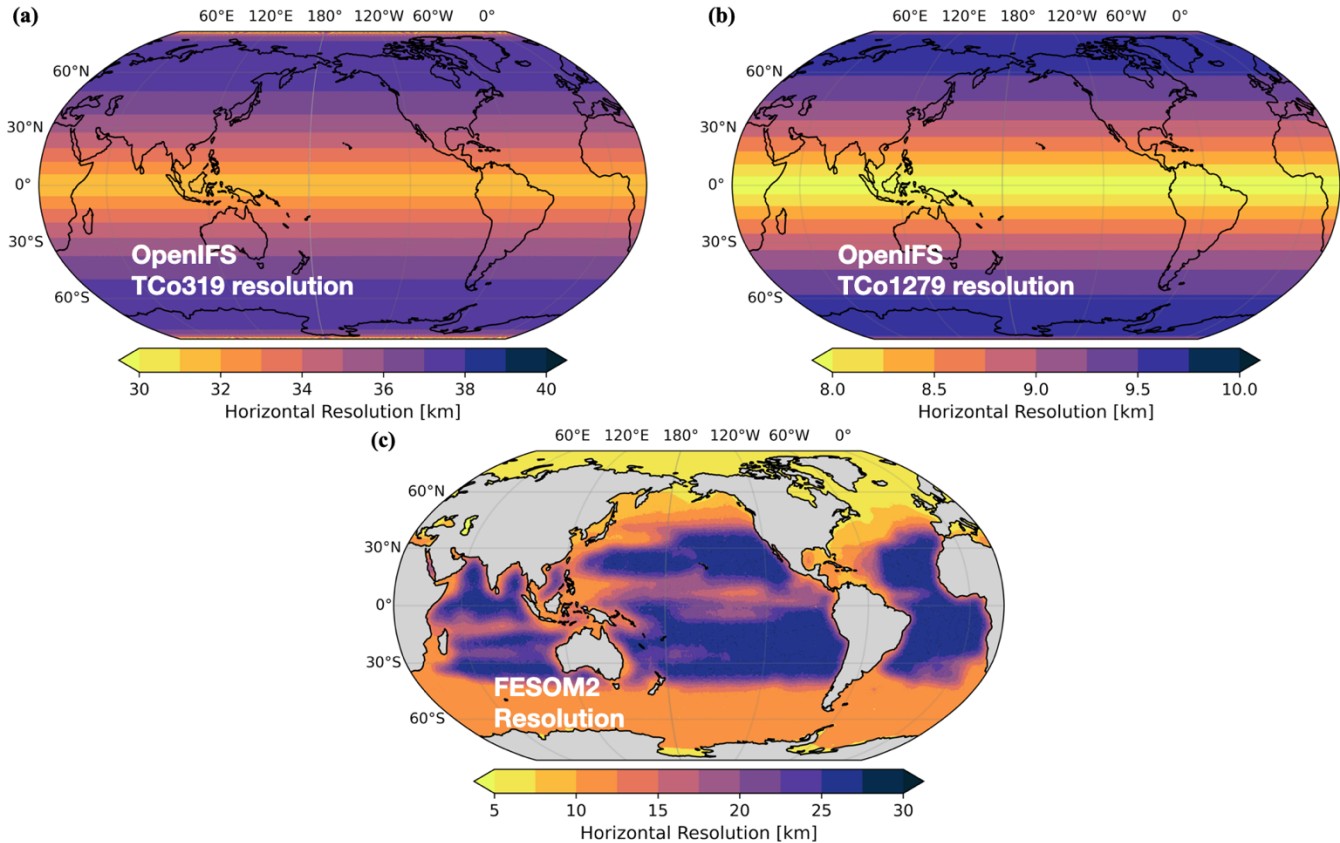

**Figure 3: Horizontal resolution of (a) TCo319 medium resolution (MR) and (b) TCo1279 high resolution (HR) configurations of the OpenIFS and the (c) FESOM2 DART mesh used for both MR and HR simulations.**

To perform HR global warming time-slice simulations extending to 2090–2100 CE, we first conducted a 100-year-long MR spin-up simulation using 1950 atmospheric greenhouse gas and aerosol conditions. This run is not further discussed here, but its end-state is used as the initial condition for our 184-year-long MR fully coupled control simulation (Fig. 4). The control run exhibits stable global mean surface temperatures of about 14 °C, which compares well with observational estimates (Hersbach et al., 2023) (Fig. 4) and shows only little radiative imbalance at the top of the atmosphere of $< \pm 0.5$ Wm$^{-2}$. The 1950 spin-up also serves as the initial condition for a transient MR scenario simulation (1950-2100), which uses historical forcings from 1950–2014 CE and greenhouse and aerosol forcing of the emission-intensive SSP5-8.5 scenario (Meinshausen et al., 2017) subsequently. The simulation reaches a $CO_2$ level of 1135 ppm by 2100 CE and the transient global mean temperature attains values of ~20.5 °C, about 6.5 °C above 1950 levels with a top of the atmosphere radiative imbalance of about 2 Wm$^{-2}$. The corresponding transient climate response (TCR) is estimated at ~ 3 °C/$CO_2$ doubling, which is somewhat outside the likely range of CMIP6 models (Meehl et al., 2020), but comparable to one of the high-end climate sensitivity



models E3SM-1-0. (Caldwell et al., 2019; Golaz et al., 2019), which exhibits an equilibrium climate sensitivity (ECS) of >

5°C/$CO_2$ doubling.

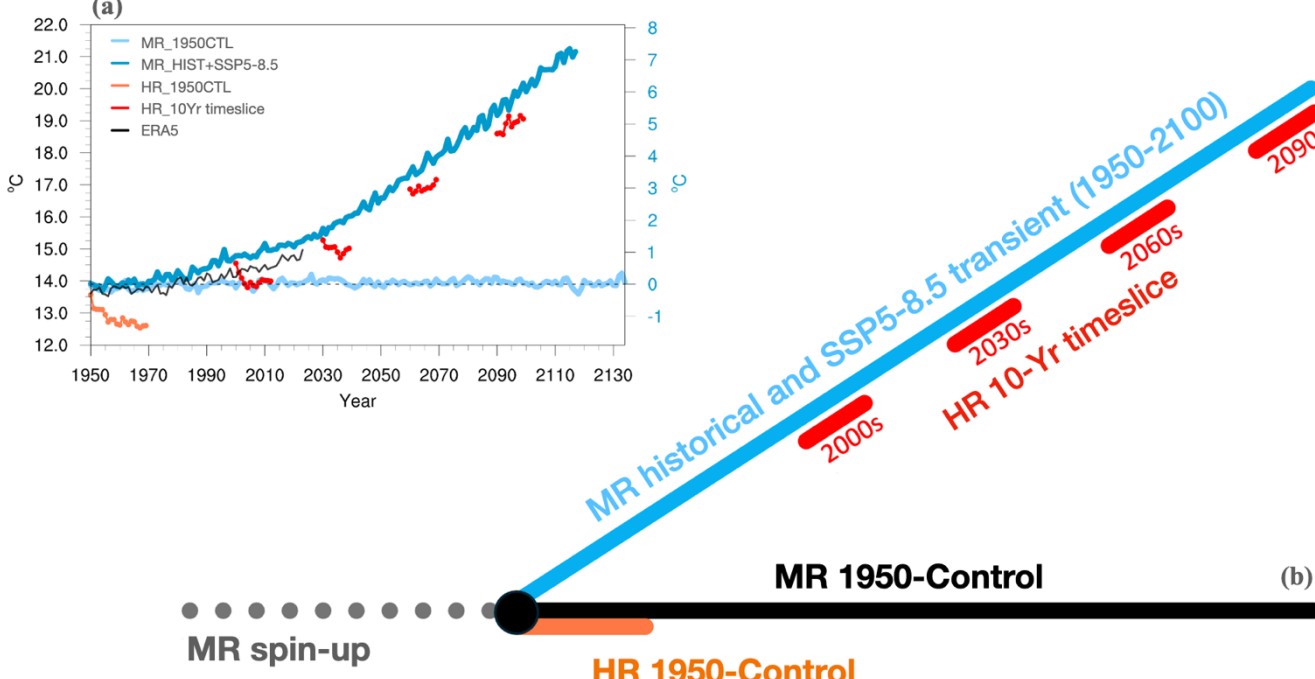

**Figure 4: Schematics of set up for transient global warming time-slice simulations along with simulated global mean surface**
**temperatures for experiments listed in Table 1: MR simulation control run (light blue), MR historical and SSP5-8.5 simulation (dark**
**blue), HR 1950 control simulation (orange), HR SSP5-8.5 transient 10-year time-slice simulation (red). Panel (a) shows the simulated**
**global mean temperatures in the simulations as well as an observational estimate from the ERA5 reanalysis (Hersbach et al., 2023).**

The 10-year HR time-slice simulations, which experience transient SSP5-8.5 forcings, are branched off from the MR SSP5-

8.5 run with ocean initial conditions corresponding to the 2000s, 2030s, 2060s and 2090s, respectively, and ERA5 atmosphere

and land initial conditions corresponding to year 1990. This atmosphere/land cold start can cause some initialization shocks,

which are partly manifested by the initial global mean surface temperature drift of the HR simulations away from the MR run

(Fig. 4a). Moreover, additional drift can occur because the coupled HR and MR simulations may have different climate

sensitivities, caused, e.g., by different cloud feedback parameters.



**Table 1: Model performance of the OpenIFS/FESOM2 model at MR (TCo319, 31 km) and HR (TCo1279, 9 km) horizontal resolution.**

| Horizontal Resolution | Cores FESOM2 | Cores OpenIFS | Cores XIOS | Simulation Years per Day | Core Hours per Year | TB per Year | Total Years | HPC |
|---|---|---|---|---|---|---|---|---|
| MR | 4800 | 1040×2 | 1×40 | 3.37 | 50,652 | 2.35 | 437 | Aleph |
| HR | 1280 | 1320×8 | 3×40 | 0.48 | 676,317 | 12.5 | (1950 EXP) 20 (2000 EXP) 13 (2090 EXP) 10 | Aleph |
| | 1280 | 4320 | 72 | 0.17 | 801,318 | | (2030 EXP) 10 (2060 EXP) 10 | GURU |

Apart from the drift, we further observe that the HR simulation is colder than the MR experiment. This offset is particularly pronounced for the 20-year-long HR control 1950 simulation, which levels off at ~1.5 °C below the MR control run (Fig. 4a). The HR time-slice simulations are also subject to transient greenhouse gas and aerosol forcings, following the SSP5-8.5 protocol and can therefore be considered short transient simulations, which will adjust over time to a transiently forced climate change trajectory.

All major components of AWI-CM3 were developed with scalability as one of the primary design criteria in mind. The base level MR simulations are performed at around 3.37 simulation years per day (SYPD) on 6920 cores of the Aleph, Cray XC-50 LC system at the Institute for Basic Science, Daejeon, South Korea. Typically, we ran several MR (TCo319L137_DART) simulations in parallel on Aleph. On the same HPC system the HR (TCo1279L137-DART) model setup with the 9 km atmosphere ran at nearly half a year per day, utilizing 11,960 cores (Table 1). The cost of the HR simulations is about 676,000 Core H/Year, compared to around 50,000 Core H/Year for the MR simulations. All simulations together (including spin-up simulations) used about 67 million CPU hours on South Korea's supercomputers Aleph and GURU. Faster MR simulations at up to 7.9 SYPD would have been possible at the cost of using the whole Aleph HPC system for one experiment. One of the major challenges of the simulations is the storage and associated analyses of the output data. A total of about 1.8 PB of output data were generated and analyzed. For OpenIFS we employed the XIOS IO server, allowing the parallelization of data output, with some on the fly in memory processing before data is written to disk for the first time. We also restricted some of the output variables for the HR simulations and used NetCDF data compression to save disk space.

## 3 Model Performance

To assess the fidelity of the MR and HR simulations, we calculate the difference between simulated temperature and precipitation fields with observational estimates (Fig. 5). Surface air temperature (SAT) in the MR simulation shows an overall warm bias, particularly over the Arctic and Southern Ocean (Fig. 5a). Such a warm bias is quite attenuated in the HR simulation, consistent with an overall lower global mean temperature (Fig. 4). There is a notable difference between the MR





07 and HR over the Arctic, related to cloud cover (Fig. S1) and sea ice (not shown). Comparing the simulated sea surface
08 temperatures (SST) in the MR and HR historical simulations with an observational climatology PHC3 (Steele et al., 2001)
09 (Fig. S1, upper right), we find an equatorial Pacific cold bias and a warm bias in the tropical Pacific and Atlantic stratus cloud
10 regions. This bias is qualitatively similar to the ensemble mean bias found in the latest CMIP models (Bock et al., 2020); but
11 the bias magnitude of the AWI-CM3 simulations is reduced considerably compared to the individual CMIP models.

Another notable reduction of the SST biases in the MR and HR historical simulations relative to CMIP models occurs in the
western boundary current regions (e.g., Kuroshio and its extension and the Gulf stream). With an ocean resolution which is
about ~$O(5{-}10)$ higher than typical CMIP models in this region (Fig. 3c), the MR and HR simulations can resolve frontal
systems more realistically, as well as mesoscale ocean eddies (Fig. 6, Supplementary Video S3) (Moon et al., 2024b), which
contribute to heat transport and recirculation. The HR and MR biases are very similar to each other. This is partly due to the
relative short 13-year integration time of the HR 2000s simulation. The SST biases (Figs. S1, S2) are also mirrored in
atmospheric 2m temperatures (Fig. 5 a, b). Over sea-ice and land the surface temperature biases relative to the 1989-2014
period in ERA5 (Hersbach et al., 2023) are most pronounced over the Barents Sea, central United States, the Sahara region,
and eastern Siberia and eastern Australia. But overall, the SST and surface air temperature of HR error is smaller compared to
MR.

In spite of some differences in SAT and SST, including slightly different global mean values, the precipitation bias patterns
(relative to the GPCP dataset) for MR and HR are almost identical (Fig. 5 c, d). This can be understood by the fact that, at least
in the tropical atmosphere, convective precipitation is largely controlled by wind convergence, which emerges in response to
SST gradients (Lindzen and Nigam, 1987). Therefore, the SST bias gradient patterns, which are very similar in MR and HR,
largely control the corresponding precipitation bias patterns. More specifically, the MR and HR rainfall bias patterns are
characterized by a relatively weak double Intertropical Convergence Zone (ITCZ) bias in the eastern tropical Pacific (Lin,
2007; Bellucci et al., 2010; Li and Xie, 2014) and a southward ITCZ bias in the Atlantic region. We also see alternating wet
and dry biases over the Maritime Continent, which need to be considered, when interpreting El Niño teleconnections in this
area (section 6).

To further illustrate the benefit of running mesoscale-resolving coupled atmosphere-ocean models on a global scale, we show
a 1-day snapshot from the HR 1950 control simulation (1969-09-10, Fig. 6). The multi-variable overlay map of SST (blue/red
shading), low cloud cover (transparent white/gray shading), hurricane wind speed (green/pink shading) and hurricane
precipitation (blue/yellow shaded inlays) shows several key features, that require high spatial resolution, including tropical
instability waves in the Pacific and Atlantic Ocean, cold ocean wakes generated by hurricanes (Chu et al., 2020), patchy
stratocumulus cloud streets in the subtropical regions, ocean eddies in the Gulf Stream region some of which were generated



by the strong wind forcing associated with bypassing Pacific and Atlantic hurricanes (see also Supplementary Video S3) (Moon et al., 2024b) and diurnal convection in the Amazon rainforest (Supplementary Video S4) (Moon et al., 2024b). The latter plays a key role for regional moisture recycling. Moreover, we see that stratus cloud bands align with SST fronts along the tropical instability waves, in a similar way as reported by Small et al. (2003). Furthermore, we observe that the high atmospheric resolution allows even for the generation of double-eye walled precipitation structures in hurricanes and the generation of strong gravity waves of the Hawaiian Islands and other topographic features (upper left inlay in Fig. 6).

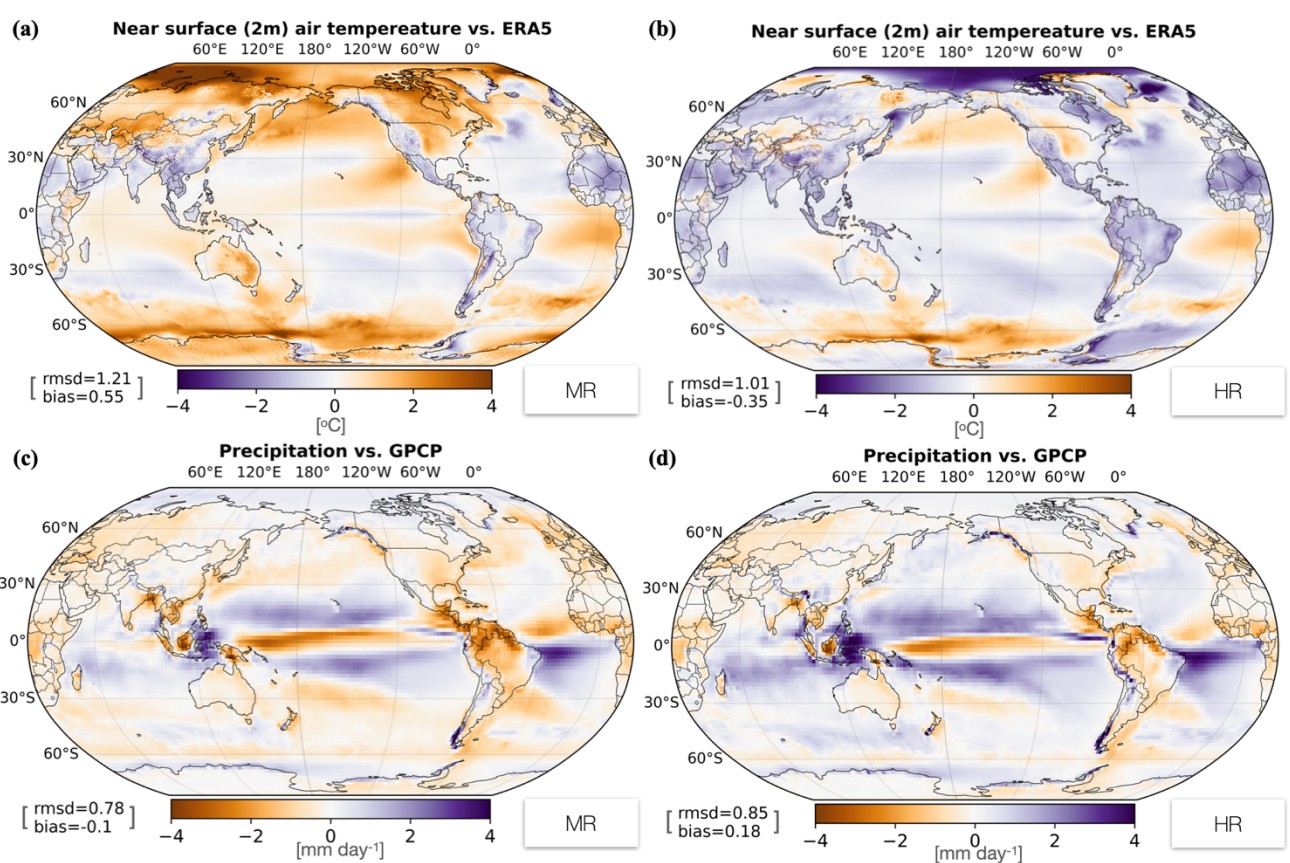

**Figure 5: Bias map for MR (a, c) and HR (b, d) historical simulations for long-term mean 2m air temperature (upper) and precipitation (lower) averaged over years 2002–2012, compared to ERA5 reanalysis and GPCP (Adler et al., 2003; Adler et al., 2012) from 1989–2014. Rmsd and bias refer to the root mean squared deviation and the global mean difference between model and reference product.**

As documented in Fig. 6, the HR simulation with AWI-CM3 resolves individual cloud systems, such as stratocumulus clouds or diurnal grid-scale convective anomalies. This will likely influence the overall performance of clouds and their response to increasing greenhouse gas concentrations. This is further illustrated in the supplementary Figure S2 (upper left), which reveals



a rather realistic representation of total cloudiness in comparison to the MODIS satellite product (Wielicki et al., 1996; Kato

et al., 2018; Trepte et al., 2019) with biases occurring mostly in the equatorial Pacific, the Arctic Ocean and over Antarctica.

It must be noted, however, that satellite cloud retrieval over Antarctica may also be subject to considerable errors.

Overall, our initial assessment of the AWI-CM3 performance demonstrates that the HR model configuration is a suitable tool

to simulate future climate at resolutions which were previously only accessible for regional climate models or SST-forced high

resolution global atmospheric models (Jung et al., 2012; Kinter et al., 2013). The relatively weak SST and precipitation biases

(Fig. 5), the very high scalability and supercomputer throughput (Table 1), as well as the versatility of our iterative modeling

protocol (Fig. 4), enable us to conduct coupled simulations of future greenhouse warming for various time periods at

unprecedented global resolutions.

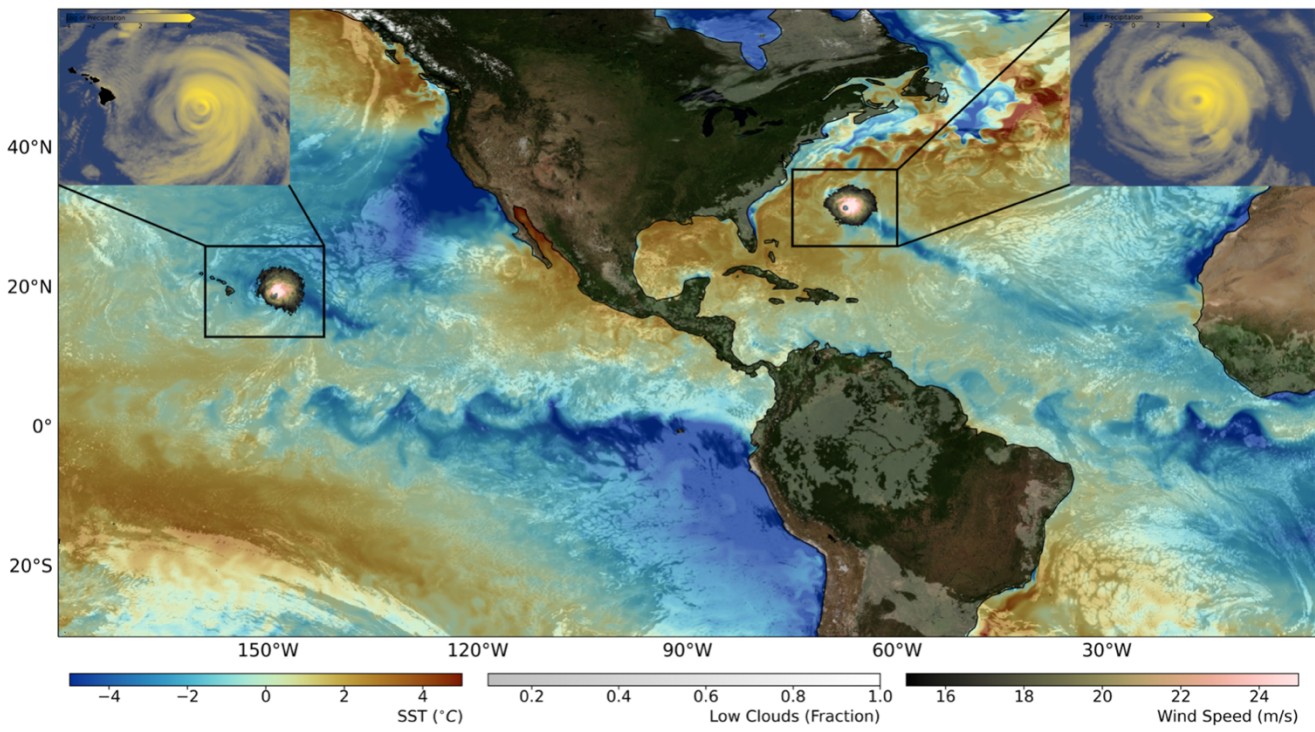

**Figure 6: Snapshot [1969-09-10] from HR AWI-CM3 20-year 1950 control model simulation showing 3-hourly data of SST (blue/red shading) minus zonal mean, low clouds (transparent gray/white shading), and 10m wind speed (green/pink shading). Hurricane precipitation (blue/yellow shaded) is shown in insets. The figure illustrates the ubiquity of mesoscale climate phenomena that can best be simulated at high spatial resolution, such as tropical instability waves in the equatorial Atlantic and Pacific, hurricanes (making landfall in Hawai'i in this simulation snapshot), ocean cold wakes generated by hurricanes, stratocumulus cloud decks and patchy day-time convection over the Amazon forest.**

574



## 4 Sensitivity to Future Climate Change

This section provides an overview of the projected mean changes in atmospheric and oceanic variables for the HR simulation. Rather than focusing on the individual time periods 2030–2039, 2060–2069 and 2090–2099 CE we normalize the responses for the transient 10-year HR simulations (relative to the 2000s present-day conditions) by their respective global mean surface temperature changes of 1.0, 2.97, 4.99 °C (relative to the 2002–2012 historical simulation), respectively (Fig. 4), and consolidate the three estimates obtained from 24 years of simulation data (last 8 years of each 10-year HR segment excluding 2 years of initial adjustment) by averaging them. This allows for a more straightforward scenario-independent interpretation of the climate change patterns and an easier comparison with experiments conducted with other climate models.

The simulations show the typical enhanced polar and land surface warming relative to the global mean (Fig. 7b). Other warming hotspots occur over the Tibetan Plateau, the Hindukush region, Northwestern and Southern Africa, the Rocky Mountains, the Andes, the Sea of Okhotsk and in the eastern Atlantic subtropical gyre. The future warming pattern is accompanied by large-scale changes of the atmospheric circulation, which are manifested in terms of a weakening of the equatorial Pacific trade winds (along with an enhanced eastern equatorial Pacific warming), a poleward shift and strengthening of the Southern Hemisphere Westerlies, which has been discussed extensively in previous studies (Cai and Cowan, 2007; Biastoch et al., 2009) and an overall intensification of Arctic surface winds (Zapponini and Goessling, 2024) (Fig. 7d). The projected relative precipitation pattern exhibits a massive intensification over the tropical Pacific, which has also been found in CMIP-type greenhouse warming simulations (Collins et al., 2010; Cai et al., 2021; Yun et al., 2021; Ying et al., 2022). However, the HR simulation differs markedly from the lower-resolution CMIP simulations (Ipcc, 2014) in that it does not show a strong relative wind speed reduction over the South Pacific subtropical regions. Moreover, in contrast to the CMIP6 multi-model ensemble mean (Ipcc, 2023), the HR simulation shows large rainfall anomalies along the southern part of East Asia, Philippine Sea, East China Sea, and toward the northern part of the North Pacific storm track path.

The representation of small-scale topography also leads to the formation of regional rainfall patterns that cannot be resolved in coarser resolution coupled general circulation models (Figs. 7e, f and Fig. 8). Striking regional features include the rainfall enhancement over the Andes, along the southern flank of the Himalayas, the Canadian Rocky Mountains, and along some topographic gradients in central and eastern Africa (e.g., Kilimanjaro) and on the eastern side of Rakhine Yoma in Myanmar. Furthermore, we observe an increased precipitation response in the northwestern North Atlantic likely due to the northward displacement of the Gulf Stream, which is dynamically consistent with the previously proposed deep atmospheric impacts of Gulf Stream temperature fronts and wind convergences (Minobe et al., 2008) (see also Fig. 7c, d). Regionally enhanced drying emerges on the western side of the land in the Mediterranean (e.g., Italy, Greece, Israel, Fig. 8) and along the mountain ranges in Central America and northeastern Brazil.





08    The simulated large-scale and regional shifts in precipitation can also be linked to projected changes in the three-dimensional

09    structure of clouds (Fig. 9). Across the different simulation snapshots, we find large-scale changes in the fraction of high (Fig.

10    9b), middle (Fig. 9d) and low (Fig. 9f) clouds.

**Figure 7: Annual mean climatology of HR simulations during 2002–2012 (left panels) and projected climate change signal**
**normalized with respect to a 1ºC global mean temperature (GMT) change (right panels) for (a, b) 2m air temperature, (c, d) 10 m**
**wind speed, and (e, f) precipitation.**

On a global scale, MR and HR simulate a global decrease in low and middle-level clouds and an increase in the upper-level

clouds (Fig. S3) in response to the SSP5-8.5 greenhouse forcing. The former contributes to an increase of incoming shortwave

radiation and the latter accelerates the greenhouse effect by emitting longwave radiation at a higher altitude, which corresponds

to a reduction in outgoing longwave radiation (OLR). Taken together, the vertical shift in clouds provides positive feedback

for greenhouse warming.

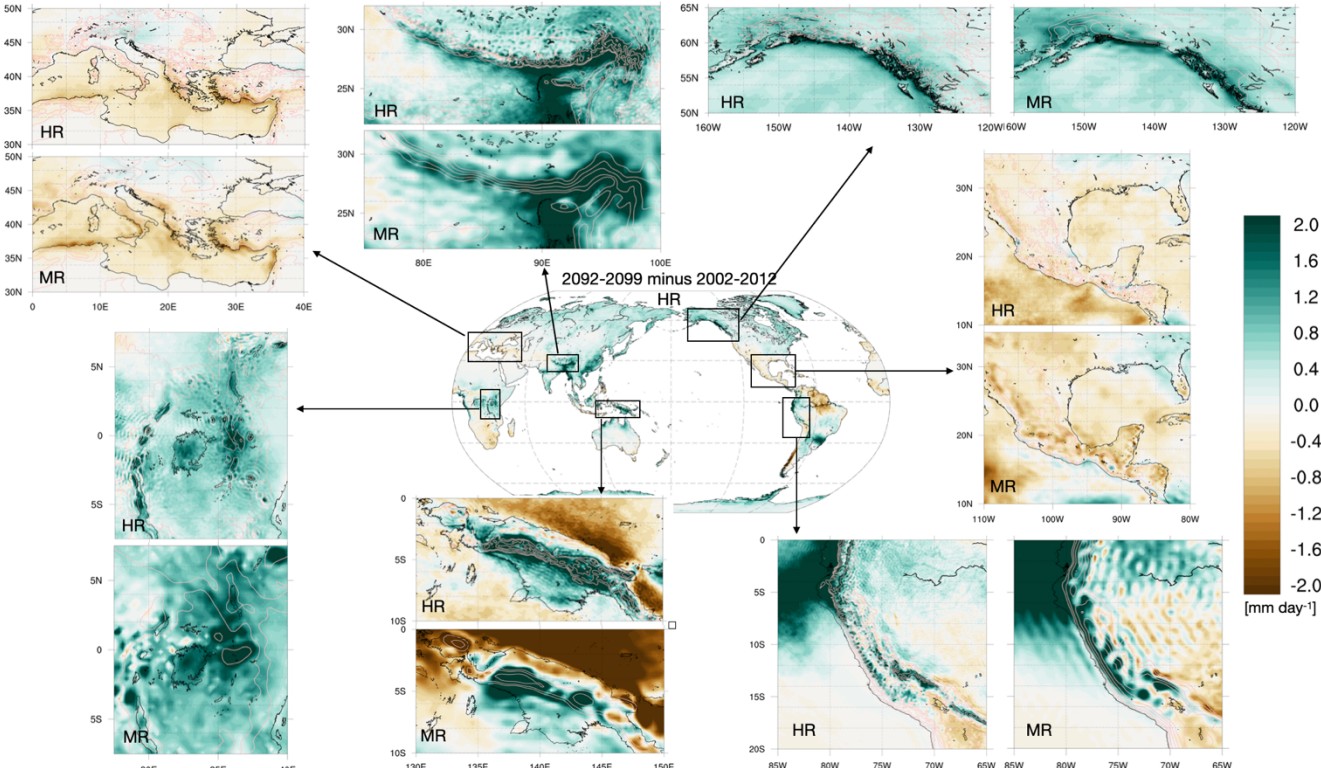

**Figure 8: Projected total precipitation changes (mm day$^{-1}$) from 2002–2012 to 2092–2099 on global (middle) and regional scales in MR and HR resolutions for (clockwise) Mediterranean, Himalaya, Pacific Northwest, Andes, Papua New Guinea, and Victoria Lake regions. The light pink lines represent the elevation of orography at 0.5, 1, 2, 3, and 4 km.**

On a more regional scale, one of the most striking features is the reduction of low and high clouds over the tropical rainforest

regions, in particular the Congo Basin and Amazonia (Fig. 9b, d). Interestingly, over central Africa this trend is not mirrored

by a corresponding regional drying trend, which hints at a much greater future rainfall efficiency (Almazroui et al., 2020).

High clouds increase in particular in the central to eastern Pacific, in polar regions, where they can also contribute to the

greenhouse effect, as well as in the western United States, South and East China Sea, and over the western Indian Ocean. The



overall structure of the cloud response is qualitatively consistent with high-ECS CMIP6 model simulations (Bock and Lauer,

2024) which further illustrates the robustness of our simulation on regional and planetary scales.



**Figure 9: Annual mean cloud climatology of HR simulations during 2002–2012 (left panels) and projected climate change signal normalized with respect to a 1ºC global mean temperature change (right panels) for (a, b) high, (c, d) middle, and (e, f) low clouds.**



To further elucidate the atmospheric circulation changes and the corresponding cloud-radiative feedback, it is necessary to
understand the underlying patterns in projected SST (Fig. 10a, b). The model simulates a strong enhanced equatorial warming
(EEW) with a slightly weaker western Pacific warming, compared to the east. This EEW pattern can be explained in terms of
the mean weak wind-speeds along the equator (Fig. 7d), the associated weak evaporative cooling (Murtugudde et al., 1996)
and the corresponding weaker Walker circulation (Fig. 10b), which further reduces eastern equatorial upwelling, thereby
contributing to the warming tendency. The EEW also leads to strong wind convergence along the equator, further weakening
of the wind speed and evaporative feedback. Additionally, it intensifies deep convection, which increases the fraction of high
clouds (Fig. 9b), enhances precipitation along the equator (Fig. 7f), and reduces surface ocean salinity (Fig. 10d), in accordance
with recent studies (Kim et al., 2023). Along with the increased temperatures, reduced salinities and equatorial wind speeds,
the mixed layer shoals (Fig 10f), which in turn also increases the response of temperatures to atmospheric heat fluxes.
Furthermore, the reduced mixed layer depth in the equatorial Pacific and the associated increased stratification lead to a
weakening of the zonal pressure gradient along the Pacific (Kim et al., 2023), which also boosts the eastern equatorial Pacific
warming.
Accelerated future ocean warming can also be found in the Barents-Kara Sea and the western side of the northern Hemispheric
subpolar gyres and along the northern edge of the Antarctic Circumpolar Current (Fig. 10b). The presence of a simulated
"warming hole" in the North Atlantic subpolar gyre is consistent with observations and model results (Caesar et al., 2018; Keil
et al., 2020) and can be linked in part to i) a reduction of the poleward heat transport due to a weakening of the Atlantic
Meridional Overturning Circulation (AMOC) (Fig. S4) and ii) a weakening of deep winter convection south of Greenland and
a regional shoaling of winter mixed layers (Fig. 10f), which increases the efficiency of winter heat-flux-driven cooling of the
ocean.





**Figure 10: Annual mean climatology of ocean variables in HR simulations during 2002–2012 (left panels) and projected climate change signal normalized with respect to a 1ºC global mean temperature change (right panels) for (a, b) sea surface temperature, (c, d) sea surface salinity and (e, f) mixed layer depth, respectively.**

The simulated differences in upper ocean salinity also reveal an overall increase in Atlantic Ocean salinity, in particular in the subtropical regions, and a reduction in the Pacific which implies that there is an increased freshwater transport from the Atlantic to the Pacific, likely through the mean trade wind export of increased water vapor concentrations from the tropical Atlantic



across the Central American Isthmus (Richter and Xie, 2010). The freshening in the subpolar North Atlantic and along the Arctic coast reflects the intensification of the hydrologic cycle in the Northern Hemisphere (Carmack et al., 2016), which increases precipitation and river runoff in high latitudes (Fig. 7f). Moreover, a weaker AMOC (Fig. S4) would also reduce the poleward salinity transport, which can lead to an accumulation of salinity (Krebs and Timmermann, 2007) in the subtropical North Atlantic and a freshening of the subpolar latitudes.

Compared to coarser resolution CMIP5 and CMIP6 models, our MR and HR simulations exhibit many more small scale current features, such as zonal jets in the subtropical oceans (Maximenko et al., 2008; Richards et al., 2006), in the Antarctic Circumpolar Current region and the Oyashio region (Fig. 11a). The high-resolution ocean model (Fig. 3c) allows us further to resolve small-scale ocean currents such as the Hawai'i Lee Counter Current west of the Big Island (Xie, 1994; Sasaki et al., 2010; Lumpkin and Flament, 2013), the Costa Rica Dome spun-up by the strong cyclonic wind-stress curl associated with the Tehuantepec and Papagayo wind-jets, which are well resolved in the HR atmosphere model (Fig. 7c), and the Galapagos wake effect of the South Equatorial Current (Eden and Timmermann, 2004) (Fig. 6).

Linked to the changes in winds (Fig. 7d) and buoyancy forcing (Fig. 10b, d) are also changes in upper ocean currents (Fig. 11b) and their instabilities and generated changes in eddies (Fig. 11d). The simulated future changes in ocean currents and eddy kinetic energy (EKE) are more prominent in regions where the currents and eddies are already energetic in the historical simulation (Fig. 11a, c). The HR model simulates a gradual slowdown of the South Equatorial Current, which is consistent with the overall weakening of the equatorial trade winds (Fig. 7d). In the Southern Ocean the northern branch of the Antarctic Circumpolar Current and the Antarctic Slope Current intensify. Furthermore, we observe a gradual strengthening and northward shift of the Kuroshio as evidenced in both the surface current and EKE, consistent with the observed trend (Yang et al., 2016). The Gulf Stream and North Atlantic Drift and the associated EKE weaken in future, which can be linked to the weakened AMOC, Gulf Stream and North Atlantic Drift in future warming climate (Fig. S4), which would be accompanied by reduced baroclinic and barotropic instabilities. Previous studies found significant correlations between the low frequency variabilities of the AMOC and EKE, suggesting a linkage also in their future changes (Beech et al., 2022). Both the Antarctic Circumpolar Current (ACC) strength and the EKE in the ACC region are projected to intensify along with the strengthening of the westerly in the future (Fig. 7d). This result indicates that the ACC is not in an eddy saturation state in the simulation. The Agulhas Return Current is projected to shift poleward, consistent with the poleward shift of the Westerlies. Both the ocean current and EKE will increase off the southwestern coast of Africa in the future, indicating an increase in Agulhas leakage and water mass transport from the Indian Ocean to the Atlantic, which has been linked previously to the poleward shift of the Southern Hemisphere Westerlies (Biastoch et al., 2009) (Fig. 7d). The Brazil and Malvinas currents show a poleward shift, which is further corroborated by observational studies in the region (Drouin et al., 2021). Overall, the projected changes in the



large-scale pattern and magnitude of ocean currents and EKE are consistent with previous modeling studies, and some inference from observations (Martínez-Moreno et al., 2021; Beech et al., 2022).

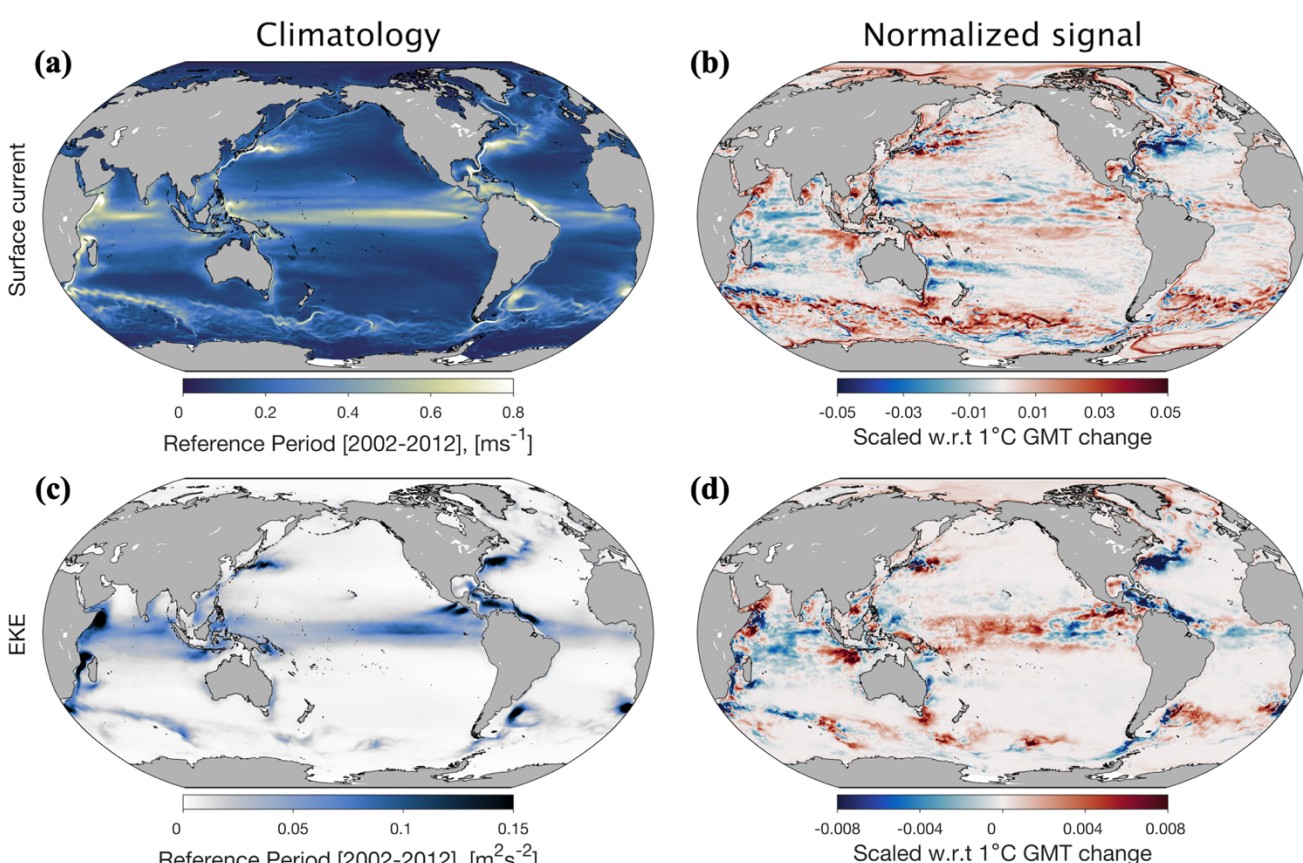

**Figure 11: Annual mean climatology of HR simulations during 2002–2012 (left panels) and projected climate change signal normalized with respect to a 1ºC global mean temperature change (right panels) for (a, b) upper ocean current speed and (c, d) eddy kinetic energy.**

When zooming into specific regions, the HR model 10-year projections provide more detailed information on regional scales, with major changes in currents and EKE occurring e.g., in the South China Sea, Lombok Strait, the Mozambique Channel.

With an ocean and sea-ice resolution of about 5–10 km in polar regions, the AWI-CM3 MR and HR simulations are ideally suited to resolve sea-ice processes realistically (Figs. S5-S6, Fig. 12). This is further illustrated in the comparison between observations of sea-ice fraction (Lavergne et al., 2019; Lavergne and Down, 2023) (shown here for September and February of 2009) in the Southern Ocean and Arctic Ocean and monthly snapshots for year 2009 in the HR simulation (Fig. 12). Even for the 2009 monthly snapshots we find an excellent qualitative agreement between observation and model simulation,



particularly in the Arctic Ocean the maximum sea-ice extent in February and minimum in September are well captured. This
is further corroborated by the simulated climatologies (Figs S5-S6). Given the high resolution of the sea-ice model, we can
identify additional features that cannot be resolved in coarse resolution models, such as coastal polynyas around Antarctica,
sea-ice eddies and large-scale cracks (e.g. Fig. 12, middle row, left and right panels, Supplementary Video S2) (Moon et al.,
2024b). As expected, the sea-ice shrinks dramatically in response to greenhouse warming and by year 2093 (Fig. 12, lower
panels, Fig. S6, left), the austral winter sea-ice around Antarctica has almost disappeared, except for a few remaining parts in
the western Weddell and Ross Seas.
In the SSP5-8.5 scenario, by the 2060s, the respective summer sea ice will disappear in both hemispheres. Similar to coarser-
resolution climate models (Notz and Community, 2020), a substantial amount of Arctic sea ice remains in wintertime in the
HR simulation. Arctic amplification of atmospheric surface temperatures (Fig. 7b) is mostly a wintertime signal associated
with winter sea-ice loss, and it can be traced back to the heat accumulation during summer due to sea-ice reduction (Chung et
al., 2021). We therefore still see a strong Arctic amplification (Fig. 7b) towards the end of the century in the HR simulation.
This effect normally disappears in Coupled General Circulation Models for even higher atmospheric $CO_2$ concentrations, once
winter sea ice disappears completely (Chung et al., 2021). The simulated HR sea-ice loss in Arctic regions is also linked with
an intensification of the hydrological cycle, more precipitation and increased middle to high cloudiness (Figs. 7f, 9b, 9d) and
a surface wind acceleration due to increased mixing of winds from aloft and reduced friction (Fig. 7d).



.36

.37

.38 **Figure 12: upper row: Monthly snapshots of sea ice fraction in observations (Lavergne et al., 2019) for year 2009 in Southern Ocean**
.39 **(September and February, left panels) and Arctic Ocean (February and September, right panels); middle row, same as upper row,**
.40 **but for the HR simulation; lower row: same as middle, but for year 2093.**

.41

.42 The oceanic response to sea-ice decline in the Southern Ocean is associated with a coastal freshening (reduced brine rejection)

.43 (Fig. 10d) and a corresponding geostrophic intensification of the Antarctic Slope Current (Fig. 11b). The sea-ice reduction

.44 further contributes to the southward shift of the Southern Hemisphere Westerlies (Russell et al., 2006) (Fig. 7d) and the

.45 associated increase of middle to high cloud cover (Fig. 9b, d) as well as precipitation near Antarctica (Fig. 7f).



## 5 Extremes

One of the key advantages of high-resolution global earth system models is the possibility to resolve mesoscale atmospheric

processes, as well as to capture the interaction between large-scale flow and small-scale topographic features, which can lead

to the generation of extreme regional climate impacts. Here we focus on the representation of atmospheric extreme events and

the projected future changes in the probability distribution of rainfall, heatwaves and tropical cyclones. To document the effect

of resolution and greenhouse warming on extreme rainfall, which is arguably one of the costliest impacts of anthropogenic

climate change, we compute the probability density functions (PDF) of the aggregated spatio-temporal rainfall data for the

periods 2002-2012 and 2092-2099 (Fig. 13a). Comparison with the ERA5 precipitation data (Hersbach et al., 2023) reveals a

slight underestimation for both the MR and HR simulations of extreme precipitation values between 25-250 mmday$^{-1}$.

Comparing MR and HR, we find that for rainfall events larger than 50 mmday$^{-1}$, the probability increases in HR by 21% for

present climate and 8% for the future climate. The climate change response is an increase in the number of extreme rainfall

events (above 50 mmday$^{-1}$) between 2002-2012 and 2092-2099 by 92% for MR and 72% for HR. This demonstrates, (i) storm

resolving models lead to stronger extreme rainfall events; a warmer climate produces more extremes, consistent with previous

studies (Rodgers et al., 2021; John et al., 2022).

In Fig. 13a, there is also an indication for different scaling behavior for accumulated precipitation up to 10 mmday$^{-1}$, between

10 mmday$^{-1}$ and 300 mmday$^{-1}$ and for larger accumulations. Similar scaling regimes have been found previously (Yang et al.,

2020) for station data in an extreme value analysis. This suggests that different mechanisms might be responsible for creating

precipitation events of different magnitudes.





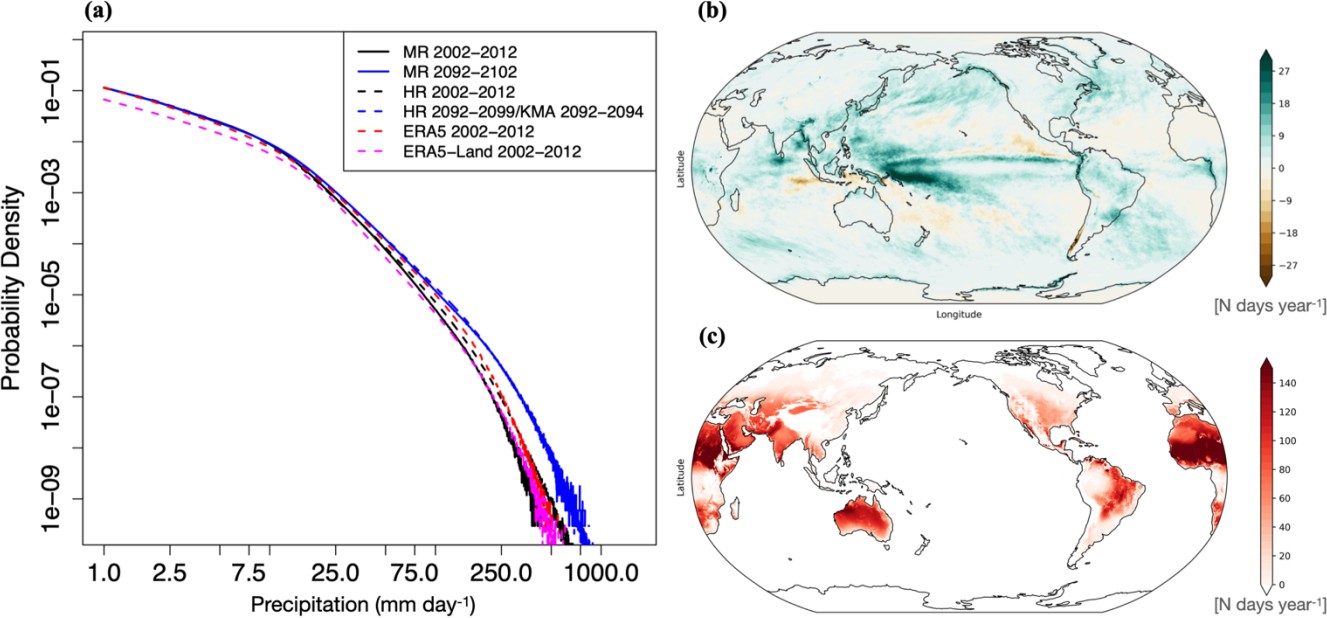

**Figure 13: a) Probability density functions for daily accumulations of precipitation. Solid lines represent MR and dashed lines represent HR; black lines are for present climate (2002-2012) and blue lines for future climate (2092-2099). Red and pink dashed lines correspond to the same analysis for ERA5 total and land precipitation data, respectively (2002-2012); b) Difference in the average number of days per year exhibiting accumulated daily precipitation of more than 50 mmday$^{-1}$, c) Difference in the average number of days when the maximum daily temperature exceeds 40 °C. The difference is computed between the time periods 2092-2099 and 2002-2012.**

Next, we examine the change in the numbers of days per year above certain impact thresholds (Fig. 13b, c). For precipitation we choose the threshold *for very heavy precipitation* (> 50 mmday$^{-1}$ based on the World Meteorological Organization definition). Consistent with previous studies (Pfahl et al., 2017; Rodgers et al., 2021), we find that the large-scale monsoon and ITCZ systems, as well as the areas affected by the major storm tracks, are projected to have more frequent very heavy rainfall events by 2090. In addition, we find an intensification in the number of extreme precipitation days across Eastern Asia (5-20 days more), Papua New Guinea, Oceania, Amazonia and central and western Africa. Moreover, the HR model simulates a strong intensification in the frequency of extreme rainfall along steep topographic slopes, e.g., Kilimanjaro, Norway, the southeastern side of the Himalayas and the Andes. These increase landslide hazards with impacts to local communities who live downslope from these extreme precipitation hotspots (Kirschbaum et al., 2020).

For the SSP5-8.5 greenhouse gas emission scenario, the number of days exceeding 40ºC maximum daily temperatures increases dramatically equatorward of 40º latitude, with some areas, e.g., Pakistan, Eastern India, parts of Amazonia, northern central Australia, northern and Southern Africa, Arabian Peninsula projected to have at least 100 days per year more (compared to present-day) exceeding this extreme temperature by 2090s. In Europe, the Mediterranean will be most affected and over



North America, California and the Great Plains region develop as hotspots. The changes over the Maritime Continent are quite small, highlighting the role of negative evaporative land-surface feedback, and the effect of reduced high clouds (Fig. 9b) which reduces regional longwave warming.

Accurate simulation of tropical cyclones is crucial for understanding their global impacts, including human and economic losses, and for assessing regional risk and preparedness. Modeling tropical cyclones presents several challenges due to their small size, their occurrence at variable distribution tails and significant variability in both time and space (see inlays in Fig. 6, Supplementary Video S5) (Moon et al., 2024b). To evaluate the model's ability in representing the global spatial distribution, annual cycle, and wind speeds associated with tropical cyclones, we employed the Okubo-Weiss-Zeta parameter (OZWP) tracking scheme, which is a resolution-independent method for detecting the genesis and tracks of tropical cyclones (Tory et al., 2013b, a). The detailed methodology can be found in the Supplementary.

One key limitation of climate models in representing tropical cyclones is the significant underestimation of maximum 10-m wind speed, even when employing models with km-scale resolution (Fig. S7). The simulation using the MR configuration fails to capture tropical cyclones with the Saffier-Simpson category 2 or higher. The same issue persists when the OWZP tracking is applied on the ERA5 reanalysis dataset, which shares a similar resolution with MR simulation. The issue of wind underestimation was previously observed during a multi-model comparison in horizontal grid spacing ranging from 250 km to 25 km, where models were found to be incapable of accurately representing highly intense surface wind speeds, while the minima of surface pressure were well represented (Roberts et al., 2020a; Roberts et al., 2020b). The HR simulation at scales of 9 km exhibits better performance compared to the MR simulation, as indicated by the peak of the PDF at the tropical storm strength (wind speeds greater than 17 ms$^{-1}$) at around 25 ms$^{-1}$, similar to observed values (Fig. S7). The secondary peak of observations at the category 4 level in the wind speed probability distribution is associated with the tropical cyclones that undergo rapid intensification, characterized by significant increases in wind speed within a short period of time (Lee et al., 2016). The MR present-day simulation fails to generate rapid intensification, whereas in HR simulation 13 cases are identified over 13 years (simulated year of 2000–2012), although their maximum wind speeds remain below category 3 levels.

In spite of the limitations in tropical cyclone wind intensity (an issue which was resolved in later cycles of the IFS model) (Bidlot et al., 2020; Majumdar et al., 2023), the HR simulation effectively captures the spatial distribution and annual cycle of tropical cyclones (Fig. 14). The total number of tropical cyclones per year in the present-day simulation is 83.8, a value that corresponds to the observed number of 83.8. In general, the model tends to overestimate tropical cyclone activities in the Western North Pacific and Southern Indian Ocean, while it underestimates them in the North Atlantic. The underestimation of Atlantic hurricane activities has been a long-standing issue observed across various models, from low-resolution CMIP5 models (Camargo, 2013) to medium resolution models (Roberts et al., 2020b), particularly when coupled with ocean models.





In our HR simulation, the underestimation may be further amplified by the negative SST bias in this area (Fig. 5, Fig. S2). The seasonal cycle of tropical cyclone frequency in each basin is well represented in the model as compared to the IBTrACS4 observational product (Knapp et al., 2010), with a large number occurring during their respective major tropical cyclone season (Fig. 14, lower panels). There is a one-month delay in their peak activity month, especially in the Eastern Pacific, whereas the North Atlantic exhibits a peak activity one month earlier than observed. Despite this one-month offset, the model generally captures the seasonal cycle of global tropical cyclone activity very well.

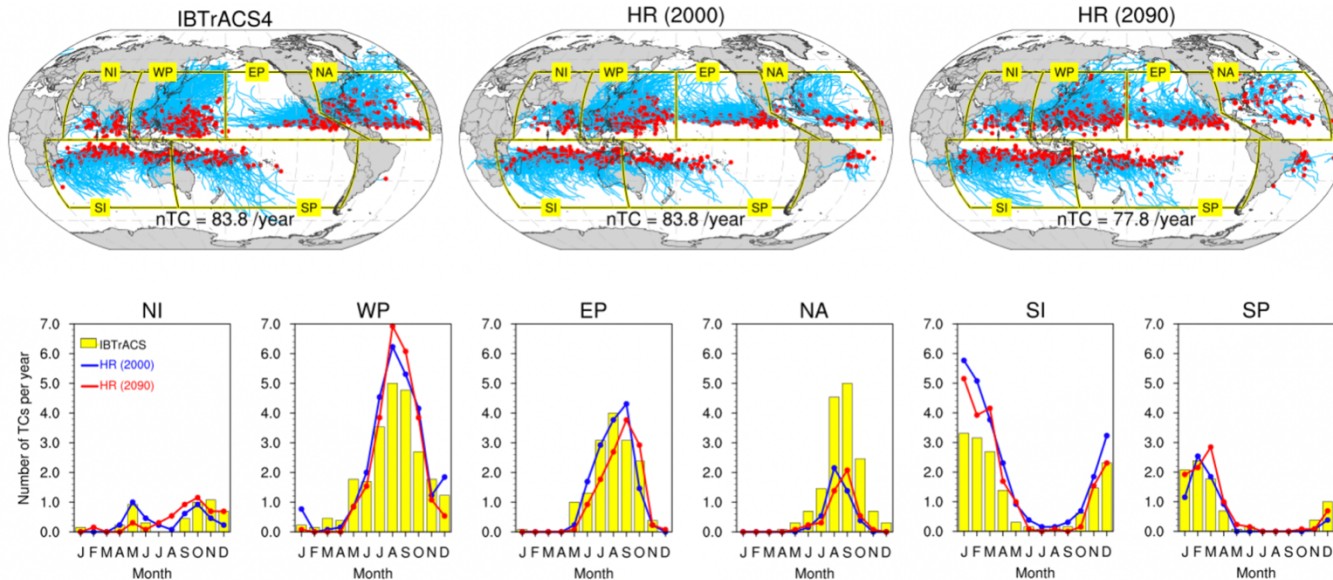

**Figure 14: Tropical cyclone tracks for present-day and future conditions: tropical cyclone tracks from (a) the observation (i.e., IBTrACS4) (years 2000-2012) and those identified by the resolution independent OWPZ tracking scheme for (b) present day simulations (13 years for HR 2000 chunk) and (c) respective future conditions (13 years for HR 2090 chunk). (d–i) Annual cycle of tropical cyclones for each basin.**

To gain further insights into the impact of greenhouse warming on tropical cyclone activity, we conducted an analysis of the 2090-2099CE HR simulation (Fig. 14). In the HR global warming simulation, the projected changes indicate a global reduction in tropical cyclone frequency by approximately 7%. This reduction in global tropical cyclone frequency primarily occurs in the Eastern Pacific, and Southern Indian Ocean. In contrast, the changes in other basins are either negligible (e.g., South Pacific) or show a slight increase (e.g., North Indian South Pacific Ocean). Interestingly, we find a zonal asymmetry across the Western Pacific, with decreased frequency in the western part and increased frequency in the eastern part of the Western Pacific (figure not shown).



Several previous studies support that the overall frequency of tropical cyclones will decrease in response to global warming, while strong tropical cyclones are likely to become more intense in future (Knutson et al., 2015; Chu et al., 2020; Roberts et al., 2020b). Our results contribute to this understanding and provide robust insights into potential changes in tropical cyclones from a high resolution coupled modeling prospective. However, further research is needed to better comprehend the impact of high-resolution grid spacing and wind speed biases on accurately representing the dynamics and structures of tropical cyclones, such as double-eye walls, rain bands etc. (Fig. 6, inlay), cyclone mergers (Supplementary Video S5) (Moon et al., 2024b) and the underlying physical mechanisms driving their future changes.

## 6 Modes of Climate Variability

Our 20-year HR control simulation and the 10-year transient greenhouse warming segments (15 years for 2090 chunk) allow us to assess the performance of important large-scale modes of climate variability, determine their regional characteristics and identify potential future changes. Here we focus on i) the MJO (Zhang, 2013), a major mode of tropical variability with a timescale of 40-90 days; ii) the North Atlantic Oscillation (NAO) (Visbeck et al., 2001; Hurrell and Deser, 2010), which plays a key role in interannual fluctuations of European winter climate; ii) the El Niño-Southern Oscillation (ENSO), known as the leading mode of interannual global climate variability (Timmermann et al., 2018). Again, here we can only give a very brief overview of some of the main projected changes in climate variability, without any in-depth mechanistic explanations. This shall be left for future studies.

### 6.1 Madden Julian Oscillation

Even though the MJO explains a considerable fraction of variance in the tropics on intraseasonal timescales, it has remained notoriously difficult to simulate in coarser resolution CMIP-type coupled general circulation models (Ahn et al., 2020; Le et al., 2021; Chen et al., 2022). The MJO shall be characterized here by the wavenumber/frequency diagram of daily OLR between 10ºS-10ºN and for the months from November to April (NDJFMA) when the MJO activity is strong. In the observations, a prominent peak occurs for eastward-propagating OLR anomalies with wavenumber 1-3 contributions and for periods of 40-90 days (Fig. 15a). The MR and HR simulations qualitatively capture these features, although the spectral peak occurs for periods of 70-80 days (Fig. 15b, c), indicating the AWI-CM3 representation of the MJO is fairly realistic compared to the majority of the CMIP5 and CMIP6 models. This encourages us to further study the response of the MJO to SSP5-8.5 greenhouse gas forcing (Fig. 15d, e). In both MR and HR future simulations (2090-2099), the MJO peak splits into two peaks. One peak characterizes low-frequency (> 90 days) OLR variability, and another one variation with a pronounced 35–40 day peak. Compared to the present-day simulations, the spectral shift towards higher frequencies is consistent with an enhancement of faster eastward-propagating signals (Chang et al., 2015).




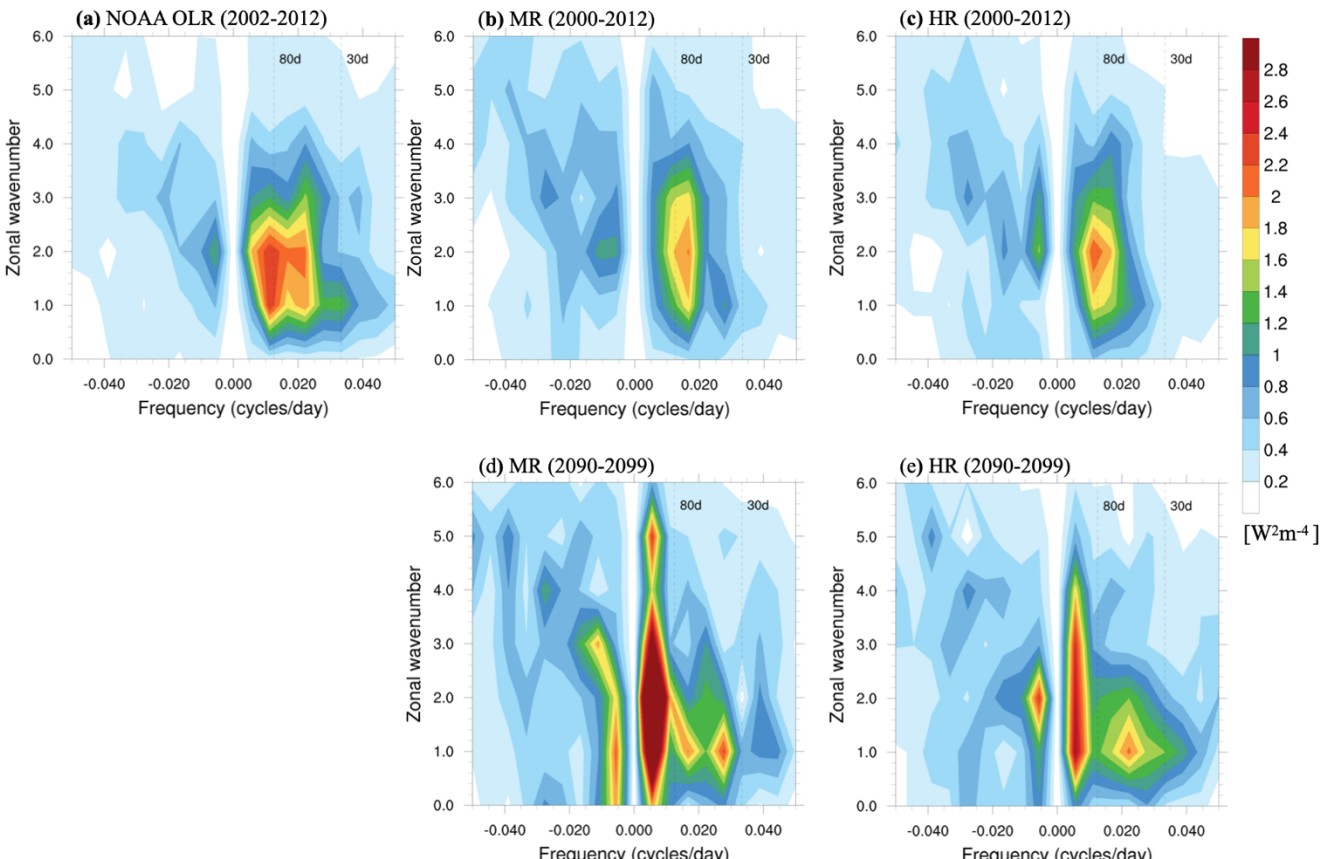

**Figure 15: Wave-number frequency diagram of Outgoing Long Wave Radiation (OLR) during NDJFMA between 10ºS-10ºN for observations (NOAA, 2000-2012) (a), MR 2000-2012 (b), HR 2000-2012 (c), MR 2090-2099 SSP5-8.5 (d), and HR 2090-2099 SSP5-8.5 (e). Frequency unit is cycles day$^{-1}$ and OLR shading is in W$^2$m$^{-4}$.**

What causes the pronounced future frequency split of the MJO mode should be investigated in more detail.

## 6.2 North Atlantic Oscillation

Year-to-year changes in winter climate over Europe with important socio-economic consequences are primarily controlled by the leading mode of atmospheric variability over the North Atlantic sector, which is commonly known as the NAO (Wanner et al., 2001; Hurrell and Deser, 2009). Here we focus on some of the regional features of the NAO simulated by MR and HR, which are usually absent in large-scale assessments of NAO impacts (Hurrell and Deser, 2009). They are particularly relevant for stakeholders (e.g. vineyards in Rhône River valley, or farmers in Georgia) and may help in mitigating negative impacts associated with NAO extreme phases (Dawson and Palmer, 2015) in sectors such as agriculture, energy generation, water management, and wildfire prevention.



Here, the NAO index is defined as the principal component (PC) timeseries of the leading Empirical Orthogonal Function (EOF) of winter (DJF) sea level pressure (SLP) anomalies over the Atlantic sector (20°-70°N, 90°W-40°E) (Hurrell and Deser, 2009). A positive NAO index is characterized by stronger westerlies over the North Atlantic and Europe, the advection of warmer marine air across Europe, drying of the Mediterranean and increased rainfall over Northern Europe (Visbeck et al., 2001). We focus on the impact on regional precipitation and wind speed, both of which are relevant for hydropower and wind-power generation (Jerez et al., 2013), as well as on surface temperature. Compared with the MR case (Fig. 16a, c), which shows the general wet-dry/windy meridional dipole across Europe for a positive NAO index, the HR simulation (Fig. 16b, d) shows topographically even more pronounced responses, in particular on the western side of Norway and Scotland (wet), the Atlas mountains (dry), the Rhône river valley (dry) and on the western side of the Mediterranean mountain ranges (drying), such as the Apennines, the Denaric Alps, Pyrenees and Pindus. This illustrates that the changes in surface westerlies and associated moisture transport with steep topography extend much further east than commonly assumed, creating important regional impacts that may be relevant for agriculture and other sectors. In the positive NAO phase, Northern Europe is more likely to experience much warmer conditions in the HR simulation compared to the MR simulation. It should be further noted that the precipitation and wind responses over Turkey, and temperature response over Portugal and Spain to the NAO phase are quite different between the MR and HR simulations.





**Figure 16: Precipitation (mmday⁻¹) regression with North Atlantic Oscillation index using 13 years (2000-2012) of the MR simulation (a) and HR simulation (b). (c) and (d), same as (a) and (b), but for the wind speed (ms⁻¹). (e) and (f), same as (a) and (b), but for the surface temperature (°C). The NAO index is based on the leading empirical orthogonal function of DJF seasonal mean sea level pressure anomalies over the North Atlantic and is normalized.**

Changes in atmospheric circulation over the North Atlantic in response to greenhouse warming may induce notable regional changes in weather and climate. For the future positive NAO phase, wet conditions on the western side of Norway and dry conditions in Portugal are apparent in both MR and HR simulations (Fig. S8) (Mckenna and Maycock, 2022).





## 6.3 El Niño-Southern Oscillation

To assess the effect of resolution (MR to HR) on the ENSO, its impacts and their projected changes, we first compare the SST anomaly standard deviation in the boreal winter season (DJF) between the observations and the different simulations (Fig. 17). Compared to the observations during the satellite era, both the MR and HR versions simulate the ENSO variability center over the equatorial central-eastern Pacific well, with slightly less ENSO SST variability in the HR control compared to the MR control (Fig. 17a, b, c). In contrast, both model versions simulate larger than observed SST variability in the extra-tropical eddy-rich regions around the Kuroshio-Oyashio Extension, the Gulf Stream, and the Agulhas. Both MR and HR simulations can reproduce the observed ENSO SST variance seasonal cycle, albeit with a notable phase bias in the MR simulations, exhibiting a seasonal minimum SST variability in early boreal summer instead of the boreal spring season seen in the observations and the HR simulations (Fig. 17f, g, h). In response to greenhouse warming, the MR configuration shows a large increase in ENSO variability in all seasons whereas the HR configuration simulates only a modest increase during the JFM season (Fig. 17d, e).

The most striking effect of different model resolutions can be seen in ENSO's global impacts at the local scale. That is, we find that increased resolution (HR vs. MR) translates into pronounced regional granularity in precipitation anomalies due to ENSO teleconnections (Fig. 18a, b, c). For instance, HR simulates a clear meridional tripolar pattern in ENSO-associated precipitation anomalies over the west coast of North America, which is absent in MR. The GPCP rainfall data do not show the tri-pole structure (Figure 18a), but the longer-term GPCC compilation of terrestrial rainfall does (not shown). Furthermore, both the positive and negative precipitation anomalies are highly amplified in HR compared to MR, especially over steep orography (Fig. 18c). This has critical implications: ENSO's impacts at the local scale are severely underestimated at model resolutions that are typically used to assess these. Hence, it is imperative to re-evaluate ENSO's various global impacts using km-scale models to provide robust climate risk assessments. Another example of the granularity of ENSO's local impact and their pronounced amplification over mountainous terrain in the HR simulation can be seen for the maritime continent region (Fig. 18d, e, f). The MR and HR simulations again exhibit very different patterns of ENSO-associated precipitation anomalies. The effect of resolution on these impacts can be demonstrated for instance by the extreme simulated drying over the high orography of the Lesser Sunda Islands (Bali, Lombok, Sumbawa, Flores, Sumba, and Timor) during El Niño in HR, whereas the same islands show slight moistening during El Niño at MR.



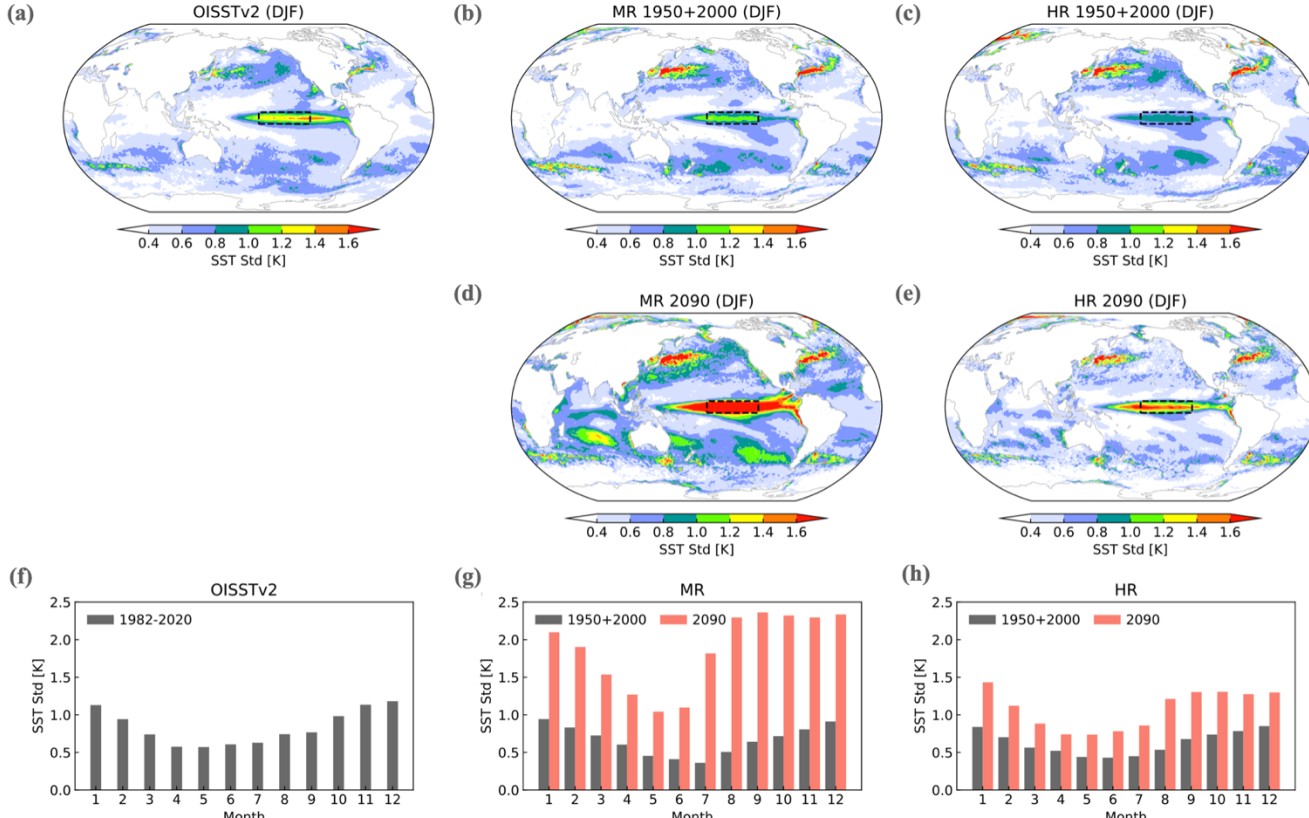

**Figure 17: El Niño-Southern Oscillation variability and its projected changes: upper row: standard deviation of DJF mean SST anomalies for observations (OISST,v2) (a), the MR 1950 control simulation (b), and the HR 1950 control + 2000 simulations [33 years in total] (c); middle row: same as upper row (b), (c), but for the 2090 time-slice [15 years in total]; lower row: seasonal standard deviation of Niño3.4 SSTA in observations (OISSTv2)(f) , MR (control, grey, 2080-2099, red)(g) , and HR (control, grey, 2090-2099, red)(h). Dashed boxes in (a)-(e) enclose the Niño3.4 region.**



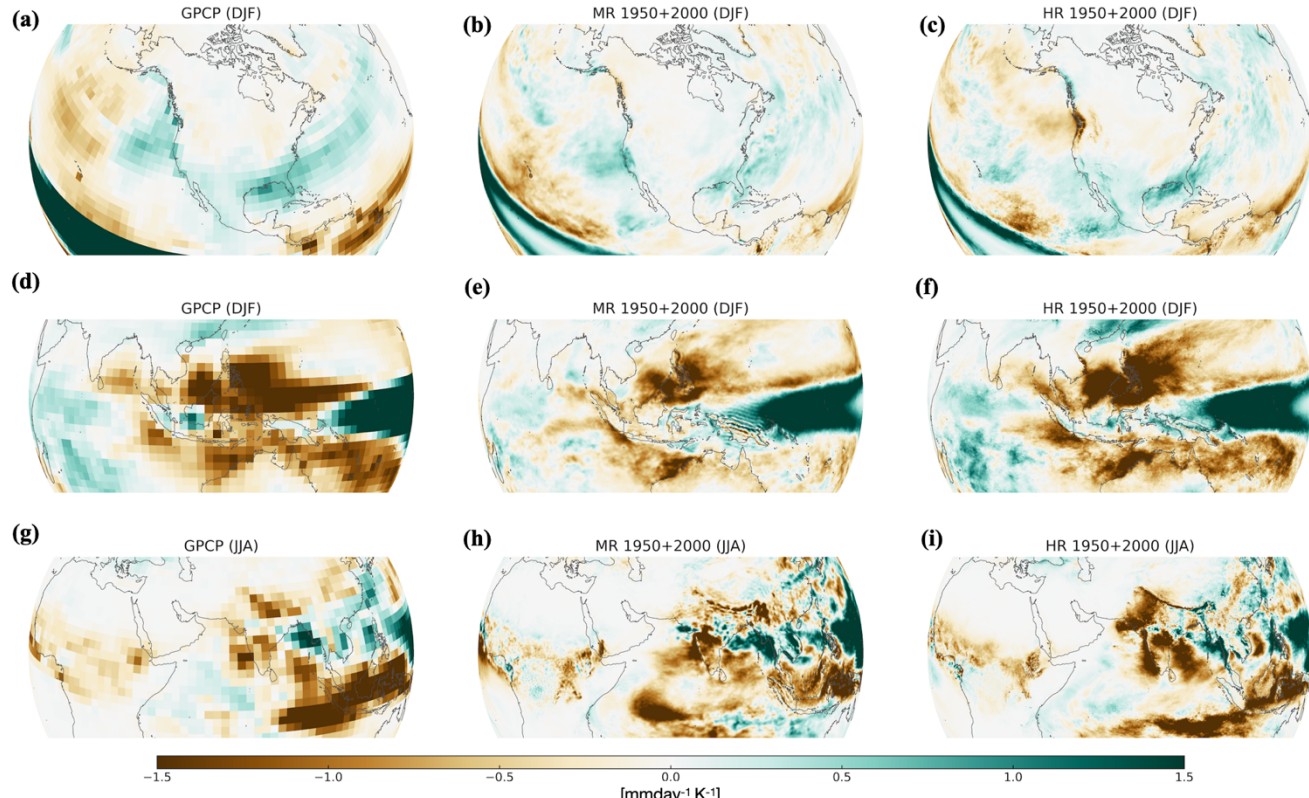

**Figure 18: Present-day observed and simulated El Niño teleconnections: Regression between DJF Niño 3.4 SST anomalies with GPCP rainfall observations over North America (a) and Maritime Continent (d) [mmday$^{-1}$°C$^{-1}$] and JJA Niño 3.4 SST anomalies with GPCP rainfall observations over Indian Ocean (g); (b),(e),(h), same as (a),(d),(g), but for 32 combined years of the MR simulation; (c),(f),(i) same as (a),(d),(g), but for 32 combined years of the 1950 and 2000 HR simulations. The data were detrended prior to the analysis.**

Future changes in ENSO teleconnections (Fig. S9, S10) indicate that on average ENSO's impact on hydroclimate anomalies is likely to intensify with greenhouse warming. This effect is particularly pronounced in DJF in the HR simulations over Europe (Fig. 19d), but also in the MR transient simulation (Fig. 19b). Under present-day conditions the ENSO linkage to DJF climate over Europe is detectable, but quite weak (Fig. 19a) (Fraedrich, 1994; Pozo-Vázquez et al., 2001; Brönnimann, 2007). Increasing temperatures, shifts in the tropospheric and stratospheric circulation and the overall enhanced availability of water vapor in a warmer climate can intensify the linkage to Europe with El Niño conditions creating wetter (drier) conditions over Southern (Northern) Europe. Future El Niño events are expected to generate a much larger tropical Pacific rainfall response (Cai et al., 2018). This in turn can strengthen the Pacific subtropical jet and its extension into the Atlantic, as well as create a troposphere-stratosphere bridge, that can influence circulation and rainfall patterns over Europe (Fereday et al., 2020). The latter is particularly pronounced in climate models that resolve stratospheric dynamics with enough vertical resolution, such as the AWI-CM3 model used here with 137 vertical levels. Changes in ENSO rainfall teleconnections are also noticeable in





other regions, such as in DJF over Southern Africa and Eastern China, Japan and Korean Peninsula, as well as in JJA over the Central Pacific, the Maritime Continent, Southern Japan and along the Andes (Fig. S9, S10), which also show a pronounced future drying in response to El Niño events.

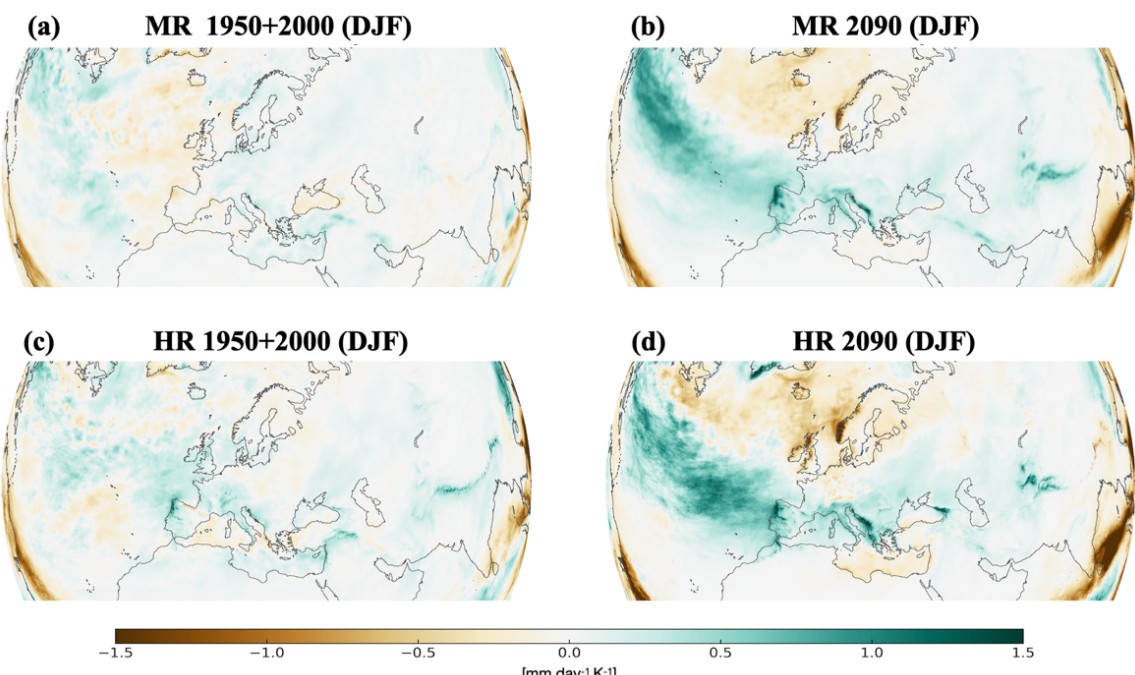

**Figure 19: Future changes in El Niño teleconnections to Europe in DJF: Regression between DJF Niño 3.4 SST anomalies with detrended simulated rainfall data over Europe [mm day$^{-1}$°C$^{-1}$]; (a),(c) for the present-day conditions (20 years of 1950 simulation and 13 years from 2000s chunk) for MR and HR resolution; (b),(d), same as (a),(c), but for 2090 CE climate conditions (using 15 years).**

## 7 Summary and Discussion

In this study, we present a new iterative method for conducting storm-resolving, fully coupled global warming simulations under various future conditions. Our approach involves a medium-resolution (MR) transient simulation with a 31 km atmospheric resolution and a 4–25 km ocean resolution, adopting the SSP5-8.5 scenario forcing. This is followed by a series of 10-year transient high-resolution (HR) time-slice simulations with a 9 km atmospheric resolution and a 4–25 km ocean resolution, initialized from the coarser transient run. The MR-HR initialization is applied only to the ocean, while the atmosphere and land are initialized with 1990 conditions. This paper compares the 31 and 9 km simulations in terms of their present-day performance and responses to increasing greenhouse gas concentrations. The present-day simulations show somewhat different background climate conditions for the MR and HR simulations. The MR simulation is, on average, 0.55°C too warm, as compared to a similar reference period in the ERA5 product. In contrast, the HR simulation is slightly cooler





than the reanalysis data (-0.35°C), illustrating that although the same ocean resolution is used, the climate background state is still affected by the atmospheric resolution. It also becomes apparent that after the initialization from the MR simulation, the HR simulation initially drifts away from the transient trajectory of the coarse-resolution model. However, this drift in global mean surface air temperature weakens in response to future warming. For SST the global mean difference between HR and an ocean reanalysis product is only 0.06°C, demonstrating the overall fidelity of the simulation. However, both simulations still show an eastern equatorial Pacific cold bias reaching a maximum of about 1.1°C, which appears to be common in CMIP5 and CMIP6 models (Li and Xie, 2014), as well as in some higher resolution simulations (Xu et al., 2022). Overall, the model shows an excellent performance in terms of mean state, but also its variability, as illustrated in the representation of tropical cyclones, the MJO, the NAO and the ENSO.

Our analysis, which covered various aspects of mesoscale to large-scale climate dynamics, highlights the additional benefit of resolving climate with local granularity in our global storm-resolving simulations. This is particularly apparent in regions of strong topographic gradients and along coastlines (Figs. 8, 16, 18), for ocean mesoscale eddies (Fig. 6, 11), and tropical cyclones (Fig. 8, 14). In fact, the 9 km global atmosphere resolution used here is similar (or even higher) than that used for many regional downscaling applications, such as those in the framework of CORDEX (Giorgi et al., 2012; Jacob et al., 2014; Giorgi and Gutowski, 2015; Gutowski et al., 2016). For regional models, data from coarser resolution CMIP-type models are typically used as lateral boundary conditions. This approach has the advantage that computing time can be saved, but also the disadvantage (depending on the domain size) that the downscaled future climate change projections often rely on model simulation is boundary conditions that cannot adequately simulate mesoscale-scale features such as tropical cyclones. Moreover, the lateral boundary conditions from the coarse resolution model are often not even a solution of the higher-resolution regional model. This can lead to spurious dynamics and transition zone effects between the internal solution and the external forcing. These effects are mitigated in our global storm resolving greenhouse warming simulations; but this comes at the expense of higher computational costs.

The new modeling protocol employed in this study has proven to be very beneficial in conducting even deeper time simulations (i.e., the end of the 21st century), without the need to run a complete multi-decadal transient scenario simulation with storm-resolving models. In the future, the storm-resolving time-slice simulations could be used to spin-off even higher resolution model simulations with the same DART ocean model set-up, but with TCo2559 (4 km) atmospheric resolution. In the midterm, the atmospheric component of AWI-CM3 will be upgraded from OpenIFS cycle 43r3 to 48r1, which brings, besides many model physics improvements and a better representation of high windspeeds, reduced precision simulations (Lang et al., 2021). The resulting reduction of computing cost and memory footprint can then be invested into even higher resolution, longer simulations or additional ensemble members.




'15 One drawback of our simulation protocol is that it induces coupling shocks at the time of initialization (Fig. 4a). Even though

'16 the same ocean initial state is used, land-surface and atmospheric variables readjust, and the background state and variability

'17 of the HR coupled system differ from the MR situation. This leads to significant model drift. Interestingly, the model drift

'18 seems to weaken with increasing greenhouse gas concentrations and global mean temperatures, which indicates an increased

'19 climate sensitivity in the HR set-up. Given that the initial years of the HR simulation chunks are not yet fully equilibrated and

'20 exhibit drift towards a new climate mean state, estimates of the transient climate sensitivity are difficult to perform. This is

'21 why we have used the HR model simulation chunks (after eliminating 2 years of spin-up time) to calculate climate change

'22 patterns normalized to 1°C global warming. This has been done by normalizing the differences between the 2030s, 2060s,

'23 2090s climate states and the 2000s chunk by their respective global mean temperature changes and averaging the 3 patterns

'24 subsequently. Assuming to first order a fast linear response to the external forcings, this procedure gives insights into the

'25 expected climate responses for a 1°C global warming on a very high regional scale. In principle this information, which is

'26 independent of the details of the greenhouse gas emission scenario, can be used for various types of planning, adaptation and

'27 management applications by scaling the normalized patterns with the expected global warming level for any given greenhouse

'28 gas emission scenario.

'29

'30 We therefore make the normalized climate change data for the HR 1°C warming projections available as NetCDF and KMZ

'31 files on the climate data server of the IBS Center for Climate Physics (https://climatedata.ibs.re.kr/data/papers/moon-et-al-

'32 2024-earth-system-dynamics-code) (Moon et al., 2024a). The scaled responses show several well-known features of

'33 greenhouse warming. These include: Increased land-ocean warming contrast, polar amplification, intensification of Southern

'34 Hemisphere Westerlies, overall increase of precipitation, except in subtropical regions and northeastern Amazonia, reduction

'35 of middle and low cloud cover and slight increase in the global mean of high clouds, shoaling of the ocean mixed layer in

'36 particular in the Southern Ocean. In addition, our HR model simulations exhibit many important regional structures that are

'37 not well captured at the scales used in many CMIP5 and CMIP6 models. Those include: Global warming hotspots in the

'38 Hindukush region, the Andes, high mountain peaks in Africa, complex terrain-specific rainfall patterns both in the mean (Fig.

'39 8) and related to the modes of climate variability (Fig. 16, 18, and Fig. S9, S10).

'40

'41 Future improvements to the modeling protocol should address the atmosphere/land cold start problem for the HR chunks.

'42 Furthermore, longer time-slice simulations might be needed to overcome model drift issues and better assess the sensitivity of

'43 modes of climate variability to greenhouse forcing. Despite these disadvantages, the modeling approach presented here allows

'44 for a relatively versatile set-up to obtain key regional climate information for future warming levels (e.g., 2090–2100), that

'45 could not be obtained easily with transient storm-resolving simulations covering the entire period from 1950–2100, due to

'46 limitations on computational resources.

'47



**Data availability.** All datasets used in the study are publicly available. AWI-CM3 MR and HR datasets are available on ICCP Climate Data Website (will be in https://ibsclimate.org after review). ERA5 reanalysis data are obtained from the Copernicus Climate Data Store: (CDS), https://doi.org/10.24381/cds.f17050d7 (Accessed on 01-FEB-2024), 2023). The data from the UHR-CESM simulations are available on the IBS Center for Climate Physics climate data server (https://climatedata.ibs.re.kr/) and upon request (https://ibsclimate.org/research/ultra-high-resolution-climate-simulation-project/). The HadISST64 can be obtained from UK Meteorological Office, Hadley Centre (https://www.metoffice.gov.uk/hadobs/hadisst/). NOAA OI SST V2 High Resolution Dataset data are provided by the NOAA PSL, Boulder, Colorado, USA, from their website at https://psl.noaa.gov. The European Centre for Medium-range Weather Forecast (ECMWF) (2011): The ERA-Interim reanalysis dataset, Copernicus Climate Change Service (C3S) is available from https://www.ecmwf.int/en/forecasts/datasets/archive-datasets/reanalysis-datasets/era-interim. The Ocean ReAnalysis System 5 (ORAS5) can be obtained from the ECMWF at https://www.ecmwf.int/en/research/climate-reanalysis/ocean-reanalysis.

**Code availability.** The Alfred Wegener Institute Climate Model AWI-CM3 is coupled atmospheric & ocean general circulation model (AOGCM) based on OpenIFS and FESOM2. OpenIFS model is available for educational and academic purposes via an OpenIFS licence (see http://www.ecmwf.int/en/research/projects/openifs). The FESOM2 model is a free software and available from Github (https://github.com/FESOM/fesom2). Code and data to reproduce the figures are available on ICCP Climate Data Website (https://climatedata.ibs.re.kr/data/papers/moon-et-al-2024-earth-system-dynamics-code) (Moon et al., 2024a).

**Supplement.** The supplement video related to this article is available online at http://doi.org/10.22741/ICCP.20240002 (Moon et al., 2024b).

**Author contributions.** AT and TJ prepared the initial manuscript and JYM led the discussion with contributions from all authors. JYM, JS, SSL, TS performed the simulations. JS, JH, TS, NK, QW, DVS, DS developed the model code changes. MAM developed the ESM-Tools. JS and WSP analyzed the model bias. JYM, EBC, RG, SSL performed projected climate signal normalized with respect to a 1°C global mean temperature change. .JPG, JC, SL, QW, GY, JYL, MZ analyzed regional aspects of mean changes. JEC and ZL analyzed tropical cyclone activity. CF and JHS performed extreme events analysis. SSL performed MJO analysis. MFS and CL performed ENSO analysis. SSL and RG performed NAO analysis. NK and SL produced and developed supplement videos. WR and SH helped with faster simulation and data access. All co-authors discussed and contributed to the final document.

**Competing interests.** Author CF is a member of the editorial board of Earth System Dynamics journal.

**Funding Financial support:** This work was supported by the Institute for Basic Science (IBS) under IBS-R028-D1. MFS was supported by NSF grant AGS-2141728. This paper is a contribution to the projects L4, S1, and S2 of the Collaborative Research Centre TRR 181 "Energy Transfers in Atmosphere and Ocean" funded by the Deutsche Forschungsgemeinschaft (DFG, German Research Foundation; project no. 274762653). DVS was supported by the Germany-Sino Joint Project (ACE, Nos. 2019YFE0125000 and 01LP2004A) and the scientific task N° FMWE-2024-0028 of MHESRF.

**Acknowledgements:** The MR transient simulations and the HR 1950, 2000, 2090 simulations were conducted on the IBS/ICCP supercomputer "Aleph," 1.43 petaflops high-performance Cray XC50-LC Skylake computing system with 18,720 processor cores, 9.59 PB storage, and 43 PB tap archive space. The 2030 and 2060 HR simulations were conducted on the supercomputer provided by the National Center for Meteorological Supercomputer (NCMS) at Korea Meteorological Administration (KMA). The team is grateful to Dr. Jang and his team from the NCMS at KMA for providing logistic and technical support. We also acknowledge the support of KREONET that enabled the fast data transfer across various platforms in South Korea. This is IPRC publication X and SOEST contribution Y. This publication is part of the EERIE project (Grant Agreement No 101081383) funded by the European Union. Views and opinions expressed are however those of the author(s)





only and do not necessarily reflect those of the European Union or the European Climate Infrastructure and Environment Executive Agency (CINEA). Neither the European Union nor the granting authority can be held responsible for them. NK benefitted from support through the project S1 of the Collaborative Research Center TRR 181 "Energy Transfers in Atmosphere and Ocean" funded by the Deutsche Forschungsgemeinschaft (DFG, German Research Foundation; project no. 274762653). The project on which this article is based also benefitted from funding by the German Federal Ministry of Education and Research of WarmWorld under the funding code Better – 01LK2202A. The responsibility for the content of this publication lies with the author.

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
