# Peer review of "Earth's future climate and its variability simulated at 9 km global resolution"

_EGUsphere, 2024_

## Referee Report (RR1)

**Review of the Manuscript egusphere-2024-2491**

**"Earth's future and its variability simulated at 9 km global resolution"**

The authors provide a very comprehensive revision of the manuscript and managed to address nearly all my earlier concerns. There is just one major point left, which is still related to the spin-up of the HR model system.

**Major comment:**

The authors state in the conclusions that *'although the same ocean resolution is used, the climate background state is still affected by the atmospheric resolution'.* This is an important point and should be mentioned earlier. To better understand the impact of the atmospheric background climate on the ocean – and thereby the coupled system - it would be good to see time series of the ocean at different model depths. This is important to get a better understanding of how long it takes the coupled model system to arrive at a new equilibrium independent of the atmospheric background state. Here, it would be sufficient to mainly focus on the 1950CTL simulations with MR and HR. How long is the ocean drifting? Do we reach an equilibrium within the 10 years? Unlike the changes OLR changes, I expect this drift to be much longer than just 2 years. This is also important to interpret the warming experiments with a transient SSP forcing, given that the ocean may still respond to the initial shock due to the atmospheric conditions.

**Minor comments:**

Page 3, line 79: 'climate models that run … require'

Page 4, line 92: replace OpenIFS/FESOM2 with AWI-CM3, as it was introduced earlier

Page 4, line 102: is the ocean in equilibrium in the two CTL simulations? See major comment.

Page 5, line 114: 'change in THE future' – in general there are many occasions where articles are missing, this should be checked throughout!

Page 5, line 128: 'via the OASIS3-MCT coupler (Ocean…) and the

Page 5, line 131: 'and up to about 38 km'

Page 5, line 140: this is not only an issue of CMIP models but all low-resolution climate models. Hence, I would remove 'participating in CMIP'.

Page 6, line 146: not a fan of including the acronyms in the figure caption. Why not adding parenthesis saying '(see Section XY for details on the individual model components)', as all acronyms are explained in this section.

Page 6, line 158: why can the stratospheric QBO not considered to be sufficiently well resolved? Please add a small explanation.

Page 7, line 167: MR exhibits stable 14C mean surface temperatures. What about HR? Is this run stable too (see earlier comment on ocean time series, maybe also adding the mean surface temperature to it).

Page 10, line 250: I would not consider more than 1C a 'slightly' different global mean temperature, given that a temperature increase of 1C in the future is major concern (in Fig. 4).

Page 24, line 489-492: How different are the precipitation values in the 1950CTL simulations of HR and MR? Are the differences similar to the historical period?

Page 37, line 763: 'neither related to the resolution of the atmosphere, nor the ocean' – are there other explanations?

Supplement:
- Again, please check the articles throughout the document.
- Fig. S5, S6, I would reverse the stippling/dashing. To grasp finer details of the significant patterns, I would suggest to stipple insignificant changes (hence, reverse the stippling).

---

## Author Response (AR2)

**Review of the Manuscript egusphere-2024-2491**
**"Earth's future and its variability simulated at 9 km global resolution"**

*The authors provide a very comprehensive revision of the manuscript and managed to address nearly all my earlier concerns. There is just one major point left, which is still related to the spin-up of the HR model system.*

: We thank the reviewer for the positive feedback on our submission. The responses to the major and minor comments are provided below. For purposes of clarity, we have put the reviewer's comments in italicized text, and responded in plain text.

**Major comment:**

*The authors state in the conclusions that 'although the same ocean resolution is used, the climate background state is still affected by the atmospheric resolution'. This is an important point and should be mentioned earlier. To better understand the impact of the atmospheric background climate on the ocean – and thereby the coupled system - it would be good to see time series of the ocean at different model depths. This is important to get a better understanding of how long it takes the coupled model system to arrive at a new equilibrium independent of the atmospheric background state. Here, it would be sufficient to mainly focus on the 1950CTL simulations with MR and HR. How long is the ocean drifting? Do we reach an equilibrium within the 10 years? Unlike the changes OLR changes, I expect this drift to be much longer than just 2 years. This is also important to interpret the warming experiments with a transient SSP forcing, given that the ocean may still respond to the initial shock due to the atmospheric conditions.*

: Thank you for the the comment regarding the ocean drifts at different model depths and its response to the atmospheric condition.

**Ocean model spin-up and its equilibrium state**

As we indicated in the model description, the MR transient simulation is conducted after 100 yrs (1850~1949) of spin-up simulation, starting from winter Polar Science Center Hydrographic Climatology (PHC3, 1900-1997 mean) (Steele et al., 2001). As shown in the figure (Fig. R1) below, the radiative imbalance (left) and vertical cold bias from the ocean surface to 250 m depth (right) are large in the first few decades and then reach near-constant radiative imbalance within the spin-up period. With respect to PHC3 climatological values, small residual drift remains at the ocean surface in the control simulation. On the other hand, the global mean ocean temperature bias in the deeper ocean persists throughout the MR control simulation. According to Streffing et al. (2022) and Rackow et al. (2018), a 3000-5000 year-long simulation would be needed to reach full equilibrium. As in Streffing et al. (2022), we extended the MR control simulation by 184 years and used as a reference for the transient simulation.

Streffing et al. (2022): AWI-CM3 coupled climate model: description and evaluation experiments for a prototype post-CMIP6 model. Geosci. Model Dev., 15, 6399–6427, 2022 https://doi.org/10.5194/gmd-15-6399-2022.

Rackow et al. (2018): Towards multi-resolution global climate modeling with ECHAM6-FESOM. Part II: climate variability, Clim. Dynam., 50, 2369–2394, https://doi.org/10.1007/s00382-016-3192-6.

Steele et al. (2001): A global ocean hydrography with a high-quality Arctic Ocean, J. Climate, 14, 2079–2087, https://doi.org/10.1175/1520- 0442(2001)014<2079:PAGOHW>2.0.CO;2.

[Figure]

Fig. R1. (a) Net radiative balance at the top-of-atmosphere (TOA) and at the surface (SFC) in the spinup + control simulation. Positive (negative) values indicate downward (upward) net heat flux. (b) Semi-logarithmic depth Hovmöller diagram of the the global mean ocean temperature bias over the spinup (model year 1850-1949) and control (model year 1950-2134) period with respect to PHC3 climatological values (Steele et al., 2001).

**Time series of the ocean temperature at different model depths**

We acknowledge the reviewer's concern regarding the model drift and its potential impact. While our high-resolution simulations offer several advantages, they also have some limitations as detailed in the manuscript. As shown in the bias map below, our HR branch-off simulation (Fig. R2b) exhibits more pronounced global ocean cooling from the surface to approximately 200 m depth over the 20-year simulation period compared to the MR branch-off simulation (Fig. R2a). This could be due to the model requiring more time to fully adjust to the initial conditions and the high-resolution atmospheric states. Achieving full equilibrium with the new high-resolution coupled model would require a significantly longer simulation period, which is practically not feasible.

However, as addressed in the manuscript, the impact of initial conditions on the atmosphere is confined to the first 1–2 years. To minimize any artificial influence from the short simulation period and the model's high sensitivity, we excluded the first two years of data, normalized the climate change patterns to 1°C of global warming, and used the 2000s (rather than the 1950s) as a reference. Additionally, this study does not examine phenomena that may be influenced by or interact with deep ocean states.

[Figure]

Fig. R2. Semi-logarithmic depth Hovmöller diagram of the global mean ocean temperature bias in the (a) MR branch off and (b) HR branch off simulations from 1950 to 1969 with respect to PHC3 climatological values (Steele et al., 2001).

**Minor comments:**

*Page 3, line 79: 'climate models that run ... require'*

: Thank you. We have revised the text.

*Page 4, line 92: replace OpenIFS/FESOM2 with AWI-CM3, as it was introduced earlier*

: Thank you. We have revised it accordingly.

*Page 4, line 102: is the ocean in equilibrium in the two CTL simulations? See major comment.*

: Thank you. Please refer the reply to the major comment.

*Page 5, line 114: 'change in THE future' – in general there are many occasions where articles are missing, this should be checked throughout!*

: Thank you very much. Our new version now adopts the simpler American English expression "in the future" without the British distinction.

*Page 5, line 128: 'via the OASIS3-MCT coupler (Ocean...) and the Page 5, line 131: 'and up to about 38 km'*

: Thank you. We have revised the text accordingly.

*Page 5, line 140: this is not only an issue of CMIP models but all low-resolution climate models. Hence, I would remove 'participating in CMIP'.*

: Thank you. We have revised it accordingly.

*Page 6, line 146: not a fan of including the acronyms in the figure caption. Why not adding parenthesis saying '(see Section XY for details on the individual model components)', as all acronyms are explained in this section.*

: Thank you. We have revised the figure caption following the reviewer's comment.

*Page 6, line 158: why can the stratospheric QBO not considered to be sufficiently well resolved? Please add a small explanation.*

: Thank you for pointing this out. This statement is to explain the vertical configuration of the atmospheric model setup, which spans from the surface to 0.01 hPa with 137 levels. This configuration is generally sufficient to resolve phenomena such as Sudden Stratospheric Warming (SSW) and the Quasi-Biennial Oscillation (QBO). SSW events and the QBO are beyond the scope of the present paper, however, we analyzed the stratospheric QBO, which the MR simulations represent with adequate amplitude but too small frequency of around 16 months.

We have revised as "In both configurations, the vertical configuration of the atmospheric model setup can be sufficient to resolve stratospheric phenomena such as sudden stratospheric warming events and the Quasi-Biennial Oscillation."

*Page 7, line 167: MR exhibits stable 14C mean surface temperatures. What about HR? Is this run stable too (see earlier comment on ocean time series, maybe also adding the mean surface temperature to it).*

: Please refer to the reply for the major comment. Additionally, the MR control simulation maintains a stable mean surface air temperature of 14°C, with a variation of 0.6°C from 1950 to 2110. The figure below (Fig. R3) presents the time series of global mean temperature (GMT at 2 m) and ocean temperature (GMOT at 2.5 m, 95 m, and 240 m depth) from the HR 1950s branch-off simulation. Over 18 years (excluding the first two years), the temperature variation in GMT at 2 m is 0.6°C, while variations in GMOT at 2.5 m, 95 m, and 240 m are 0.4°C, 0.3°C, and 0.3°C, respectively.

[Figure]

Fig. R3. Time series of the global mean temperature at 2 m (orange) and the ocean temperature at 2.5 m (sky), 95 m (light blue), and 240 m (dark blue) depth from the HR branch-off simulation (1950–1969).

*Page 10, line 250: I would not consider more than 1C a 'slightly' different global mean temperature, given that a temperature increase of 1C in the future is major concern (in Fig. 4).*

: Thank you for the comment. We have removed 'slightly' in the sentence.

*Page 24, line 489-492: How different are the precipitation values in the 1950CTL simulations of HR and MR? Are the differences similar to the historical period?*

: The global mean bias of precipitation is similar. The global mean precipitation difference between MR and HR in 1950s (2000s) is 0.02 (0.01) mm/day.

*Page 37, line 763: 'neither related to the resolution of the atmosphere, nor the ocean' – are there other explanations?*

Thank you for pointing this out. We have revised the text as follows:

"Apparently, their presence is neither related to the resolution of the atmosphere, nor the ocean model. Model biases may also stem from other factors, such as parameterizations, initial conditions, or limitations in model physics and the representation of feedbacks. Further investigation is needed for a better understanding."

Supplement:

- *Again, please check the articles throughout the document.*

: Thank you. We have carefully revised the manuscript.

*- Fig. S5, S6, I would reverse the stippling/dashing. To grasp finer details of the significant patterns, I would suggest to stipple insignificant changes (hence, reverse the stippling).*

: Thank you. We have revised the figure as shown below.

[Figure]

**Editor comment**

*Ja-Yeon Moon et al. present a new framework for simulations with a very high-resolution coupled atmosphere-ocean circulation model. This draws on two versions of AWI-CM3, with the same high-resolution ocean grid, and a medium (31km, MR) and a high (9km, HR) atmospheric grid, where the HR simulations are initialized from an MR ocean state, and with 1990CE land surface conditions. The authors place the two model versions for a control (1950-1970), and the historical (2002-2012) results in the context of CMIP6 results in terms of regional variability, precipitation extremes, and modes of variability. They provide results from MR and HR simulations for three future decades simulated along RCP8.5 (starting 2030, 2060, 2090). The HR simulations show initialization shocks, and for 2030 and 2060 global mean trends that differ from that in the MR simulations. Based on the two reviewer's assessments, and my own reading, I conclude that the revised version of the manuscript is substantially improved compared to the initial submission, in particular with regards to the figures, following reviewer recommendations. The manuscript is interesting for the community for structural aspects of model building, presenting the data, and initial results indicate that they could substantially contribute to a better understanding of the physical climate system. The authors improved their discussion of the initialization shocks and the model drift, with sensitivity experiments using the MR version of the model showing that the land surface conditions are not the primary cause of the decadal cooling trend for the HR model. The reviews also indicate that the manuscript does, still fall short on several aspects, which should be fixed prior to acceptance. In preparing your revised version of the manuscript, please take into account in particular the following points:*

: We thank the editor for the positive outlook on our submission and constructive comments. We provide below our replies to the specific comments and the edits that implemented in the manuscript as a result of these. For purposes of clarity, we have put the editor's comments in *italicized* text, and responded in plain text.

*a) Consider the detailed comments of Reviewer 2 in the revision, which center around improved discussion of the model drift, and ask for minor clarifications throughout the text.*

: Thank you for the comment. We provided detailed explanations to address the reviewer's comments as thoroughly as possible. Please refer to the reply to the reviewer.

*b) Given that the land surface initialization is not the main cause of the cold drift, please mention briefly, in the manuscript the reasons for using 1990 land surface conditions the restarts (e.g., for technical reasons).*

: In Section 2 (Model Description and Experiment Setup), we have revised the content to address reviewer-1's previous comment and the editor's feedback concerning the initial conditions for the year 1990, as described below:

"The 10-year-long HR time-slice simulations, which are driven by transient SSP5-8.5 forcings, are branched off from the MR SSP5-8.5 run with ocean initial conditions corresponding to 2000, 2030, 2060 and 2090 CE, respectively. The atmospheric and land initial conditions for each of these time slices are not

taken from the MR simulation (due to technical implementation difficulties), but rather from archived data from ECMWF for the OpenIFS model, version v43r3, which correspond to the year 1990 ([https://confluence.ecmwf.int/display/OIFS/6.2+OpenIFS+Input+Files](https://confluence.ecmwf.int/display/OIFS/6.2+OpenIFS+Input+Files))."

*c) Statistical measures used to assess significance in many plots are now mentioned. Please include your strategy for the assessment of the degrees of freedom used in the statistical tests (i.e. no autocorrelation in space/time). Is this based on default settings in which software? The maximal number of degrees of freedom for a high-resolution simulation would always increase, but is limited by inherent spatial and temporal autocorrelation of the simulated climate (f. ex.: Wang, Xiaochun, and Samuel S. Shen. "Estimation of spatial degrees of freedom of a climate field." Journal of climate 12.5 (1999): 1280-1291. [https://doi.org/10.1175/1520-0442(1999)012](https://doi.org/10.1175/1520-0442(1999)012)<1280:EOSDOF>2.0.CO;2 ). While it is nontrivial to determine the correct degrees of freedom, it is relevant for the interpretation and comparison to clarify the choices used in the assessment, either in a summary statement (e.g., "we used default setting in the "stats" package in version X of software Y") or where you mention the statistical tests (e.g., assuming 3650 degrees of freedom for each 10-year time series of daily precipitation at the gridpoint level).*

: Following the comment, we have added clarifications on the statistical assessment in Figs. 5, 7, 9, 10, 11, 13, 18, 19, S3, S4, and S14 of the revised manuscript, explaining the methodological choices. Figs. S5 and S6 from the previous supplementary have been replaced by Figs. 7, 9, 10, and 11 in the revised manuscript, which now include significant regions.

The degree of freedom (DOF) differs across the figures:

- Fig. 5: Based on 11-yr (2002-2012) annual mean data.

- Figs. 7, 9, 10, and 11: Based on 288 (8-year periods from the 2030s, 2060s, and 2090s) monthly data vs 132 reference monthly data.

- Fig. 13a: Uses a combination of daily and horizonal grid samples.

- Fig. 13b: Uses 9-yr (2092-99) vs 11-yr (2002-2012) reference data.

- Figs. 18, 19, and S16: Based on 32-yr combined samples.

For further details, please refer to the figure captions in the revised manuscript.

*d) Tone down the language with regards to the superiority of the HR model for climate projections, given the initial assessment in this paper (e.g., line 366), and for the realism of ENSO in HR vs. MR (l929) which is inconsistent with the results section.*

: Thank you for the comment. We have made efforts to ensure that our statements remain balanced and avoid overly strong wording throughout the text.

*e) Enhance the discussion on limitations and potentials of the simulations (as they are) and for the future development of the simulation framework, considering model drift despite apparent TOA balance and CO2 increase, challenges around spinning up and potentially re-tuning high-resolution model versions, or coping with different model sensitivities as a task for future work. The author replies clearly show using the*

*MR simulations that the land surface conditions are not the primary reasons for the cold drift, but this opens up new questions with regards to the climate sensitivity and disequilibrium effects in the HR simulations that may bias or increase variance of the simulated climate. A future-work-oriented final paragraph could include this nicely, and outline opportunities for the research community, as well as necessary future work.*

: We appreciate the detailed comment on the current simulation framework and the important challenges. We have thoroughly revised the discussion section in response to the provided comment and have added/modified the last paragraph as:

"Future improvements to the modeling protocol should include longer time-slice simulations to overcome model drift issues due to different climate sensitivities between MR and HR. Other factors that may contribute to the initial model drift and the discrepancies between MR and HR projections include the representation cloud feedbacks and atmosphere-ocean feedbacks. Additionally, HR model simulations typically require more time and computational resources to reach equilibrium, making spin-up and re-tuning significant challenges. Future work should focus on gaining a deeper understanding of the factors which contribute to model drift and developing more efficient optimization and model calibration methods to address these challenges. Despite these disadvantages, the modeling approach presented here allows for a relatively versatile set-up to obtain key regional climate information for future warming levels (e.g., 2090–2100), that could otherwise not be obtained easily with transient climate model simulations covering the entire period from 1950–2100."